

# Time-dependent Schwinger boson mean-field theory of supermagnonic propagation in 2D antiferromagnets

**Martijn D. Bouman and Johan H. Mentink**⋆

Radboud University, Institute for Molecules and Materials (IMM),
Heyendaalseweg 135, 6525 AJ Nijmegen, The Netherlands

⋆ j.mentink@science.ru.nl

## Abstract

Understanding the speed limits for the propagation of magnons is of key importance for the development of ultrafast spintronics and magnonics. Recently, it was predicted that in 2D antiferromagnets, spin correlations can propagate faster than the highest magnon velocity. Here we gain deeper understanding of this supermagnonic effect based on time-dependent Schwinger boson mean-field theory. We find that the supermagnonic effect is determined by the competition between propagating magnons and a localized quasi-bound state, which is tunable by lattice coordination and quantum spin value $S$, suggesting a new scenario to enhance magnon propagation.



# 1 Introduction

Studying the propagation of coherent magnons with the highest possible frequencies and the shortest possible wave lengths is considered as one of the main challenges in ultrafast spintronics and magnonics [1–3]. Antiferromagnets are ideal candidates to investigate such high frequency magnons since their natural frequencies are in the THz range, the highest of all magnets. Combined with intrinsically low dissipation, antiferromagnets are promising for high-speed low-energy data processing [4–8]. So far, most efforts towards short wavelength coherent magnons rely on effectively reducing the excitation volume [9, 10]. However, in this case the vast majority of magnons generated still have wavevectors close to the center of Brillouin zone. A promising approach to bypass this problem is to study pairs of counterpropagating magnons, such that their total momentum vanishes. In antiferromagnets, an efficient strategy to accomplish this is via the optical perturbation of exchange interactions. This mechanism is well-studied in spontaneous Raman scattering [11, 12] and generates pairs of magnons across the whole Brillouin zone. Together they form the two-magnon mode which peaks near the frequency of zone-edge magnons. Importantly, the same mechanism was also found effective for the generation of coherent dynamics of the two-magnon mode with ultrashort laser pulses [13, 14].

From a theoretical point of view, the dynamics of magnon pairs differs significantly from that of single magnons. First of all, by definition magnon-pair dynamics goes beyond a single-particle description. Moreover, since the two-magnon mode is dominated by short-wavelength magnons, a classical long-wavelength continuum theory is expected to fail. For example, within continuum theory one expects that the propagation speed is limited by the magnon group velocity [15], which may no longer be the case at short wavelengths. Moreover, the quantum nature of the spins may be of key importance, leading to completely different selection rules for the excitation of magnon pairs than expected from classical theory [16, 17]. On top of that, quantum effects enhance the interactions between magnons beyond what is obtained in classical interacting spin-wave theory. For simple canonical models such as the square lattice Heisenberg model in two dimensions (2D), non-trivial quantum corrections even arise for the single-particle magnon spectrum at short wavelenghs [18–22]. Moreover, the 2D antiferromagnetic Heisenberg model features an exceptionally broad two-magnon spectrum [23–25]. However, rather little is known about the quantum effects and magnon-magnon interactions for the propagation of magnon-pairs.

A first theoretical study on the quantum effects of magnon-pair propagation was recently conducted based on neural network quantum states (NQS) [26]. Beyond the limitations of existing methods [27], this approach offers a practically unbiased solution to the full quantum spin dynamics and was found highly efficient for studying the antiferromagnetic Heisenberg model in 2D [28]. Interestingly, this study predicted a regime of supermagnonic propagation, where spin correlations transiently propagate with a velocity that is higher than the magnon group and phase velocity. To asses the origin of the of the supermagnonic effect, it is natural to compare to analytical results on magnon-pair propagation. In [28], such an effort was taken based on Schwinger boson mean-field theory (SBMFT). This provides a solution of the 2D Heisenberg model for the $SU(n)$ generalized spin operators, which becomes exact in the limit $n \to \infty$. Furthermore, SBMFT shows good qualitative agreement with the results from neural network quantum states, and the dependence of the supermagnonic effect on the quantum spin value $S$ showed that the supermagnonic effect stems from quantum fluctuations. It will be very interesting to further investigate the role of quantum fluctuations to possibly enhance the supermagnonic effect. This can be done not only by changing $S$ but also by changing the lattice coordination $z$ and may possibly lead to new mechanisms for tuning the propagation of spin correlations in antiferromagnets.

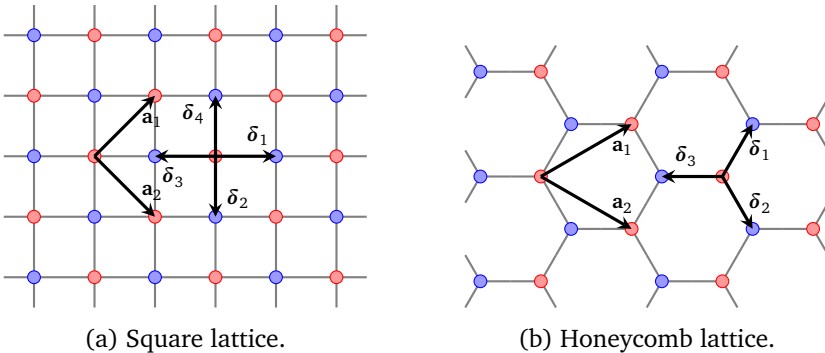

(a) Square lattice.       (b) Honeycomb lattice.

Figure 1: Diagram of the lattices in consideration, with the lattice vectors $\mathbf{a}_i$ and nearest neighbor vectors $\boldsymbol{\delta}_i$.

Being distinct from both the spatial mean-field approximation (which becomes exact for $z \to \infty$) and the semiclassical large $S$ expansion (which becomes exact for $S \to \infty$), SBMFT allows for a systematic analysis of quantum effects arising from both the discrete lattice and the discrete nature of the quantum spin. Moreover, given the good qualitative agreement between NQS and SBMFT, the latter is a good starting point for gaining insight in the effect of different quantum fluctuations on supermagnonic propagation. Such a study, however, has not been attempted so far.

The main aim of this paper is to obtain a deeper understanding of the supermagnonic effect, by investigating the contribution of different quantum effects. To this end, we first present a detailed derivation of SBMFT for space-time dynamics in the linear response regime, along with the results of linear spin-wave theory (LSWT) to quantify the regime of the supermagnonic effect. Second, we apply this method to the dynamics of spin correlations on the 2D antiferromagnetic Heisenberg model on both the square and honeycomb lattice. Interestingly, we identify that the significance of the supermagnonic effect is not solely controlled by the amount of the quantum fluctuations, but results from the competition between a localized quasi-bound state and quasi-noninteracting propagating magnon-pairs. This implies a new scenario to control magnon-propagation at the shortest space and time scales in antiferromagnets.

The remainder of the paper is organized as follows. First, we discuss the static SBMFT for the square and honeycomb lattice and subsequently the time-dependent linear response framework. This yields time-dependent self-consistency equations that we solve analytically using the Laplace transformation, allowing us to evaluate the space-time dynamics of spin correlations. Next, we apply this framework to study the propagation of spin correlations after ultrashort perturbation of exchange interactions that trigger dynamics of magnon-pairs. We analyze the results by comparing SBMFT with LSWT and finally draw conclusions on the origin of the supermagnonic effect. Three appendices are included to provide additional details on LSWT and on SBMFT for static and dynamic perturbations.

## 2   Static SBMFT

We consider the bipartite antiferromagnetic Heisenberg model given by

$$\hat{H}_0 = J \sum_{\mathbf{r} \in A} \sum_{\boldsymbol{\delta}} \hat{\mathbf{S}}_{\mathbf{r}} \cdot \hat{\mathbf{S}}_{\mathbf{r}+\boldsymbol{\delta}} \,, \tag{1}$$

on the square and honeycomb lattice (see Fig. 1), where $J$ is the exchange interaction, and with $N = 2L^2$ number of spins. All derivations in this paper will be done for arbitrary temperature $T$,

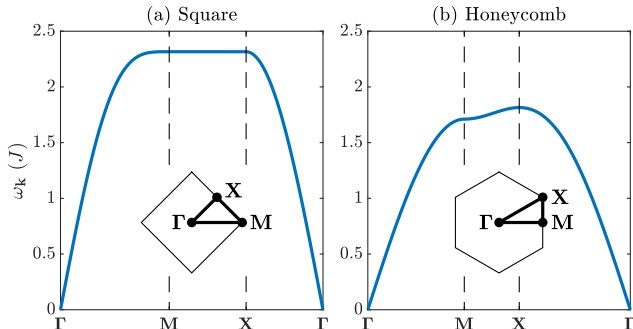

Figure 2: Dispersion relation for (a) the square and (b) the honeycomb lattice along high-symmetry paths in the Brillouin zone (inset).

but in evaluation and plotting we will restrict ourselves to $T = 0$. In the Schwinger boson representation, the spin operators are described by bosons with two 'flavors' indicated by the label $s = \pm 1/2$. Furthermore, we define different boson operators on the different sublattices. The Schwinger boson mapping is given by [29]

$$
\begin{aligned}
&\text{for } \mathbf{r} \in A, && \text{for } \mathbf{r} \in B, \\
&\hat{S}_{\mathbf{r}}^{+} = \hat{a}_{\mathbf{r},\frac{1}{2}}^{\dagger} \hat{a}_{\mathbf{r},-\frac{1}{2}}, && \hat{S}_{\mathbf{r}}^{+} = -\hat{b}_{\mathbf{r},-\frac{1}{2}}^{\dagger} \hat{b}_{\mathbf{r},\frac{1}{2}}, \\
&\hat{S}_{\mathbf{r}}^{-} = \hat{a}_{\mathbf{r},-\frac{1}{2}}^{\dagger} \hat{a}_{\mathbf{r},\frac{1}{2}}, && \hat{S}_{\mathbf{r}}^{-} = -\hat{b}_{\mathbf{r},\frac{1}{2}}^{\dagger} \hat{b}_{\mathbf{r},-\frac{1}{2}}, \\
&\hat{S}_{\mathbf{r}}^{z} = \frac{1}{2}\left(\hat{a}_{\mathbf{r},\frac{1}{2}}^{\dagger} \hat{a}_{\mathbf{r},\frac{1}{2}} - \hat{a}_{\mathbf{r},-\frac{1}{2}}^{\dagger} \hat{a}_{\mathbf{r},-\frac{1}{2}}\right), && \hat{S}_{\mathbf{r}}^{z} = -\frac{1}{2}\left(\hat{b}_{\mathbf{r},\frac{1}{2}}^{\dagger} \hat{b}_{\mathbf{r},\frac{1}{2}} - \hat{b}_{\mathbf{r},-\frac{1}{2}}^{\dagger} \hat{b}_{\mathbf{r},-\frac{1}{2}}\right),
\end{aligned}
\tag{2}
$$

supplemented with the constraint $\sum_{s} \hat{a}_{\mathbf{r},s}^{\dagger} \hat{a}_{\mathbf{r},s} = \sum_{s} \hat{b}_{\mathbf{r},s}^{\dagger} \hat{b}_{\mathbf{r},s} = 2S$, which is enforced in the Hamiltonian by a Lagrange multiplier $\lambda_0$. By defining the bond operator $\hat{A}_{\mathbf{r},\mathbf{r}'} = \sum_{s} \hat{a}_{\mathbf{r},s} \hat{b}_{\mathbf{r}',s}$, we can write $\hat{\mathbf{S}}_{\mathbf{r}} \cdot \hat{\mathbf{S}}_{\mathbf{r}+\delta} = S^2 - \frac{1}{2}\hat{A}_{\mathbf{r},\mathbf{r}+\delta}^{\dagger} \hat{A}_{\mathbf{r},\mathbf{r}+\delta}$. This way, the Hamiltonian can be generalized from 2 to $n$ number of flavors, and has SU($n$) symmetry [29, 30]. The SBMFT is obtained by performing a mean-field expansion in the bond operator (which is exact in the limit $n \to \infty$), and taking $n = 2$ [29–31]. This is equivalent to the leading order result in a formal $1/n$ expansion [32]. In $\mathbf{k}$-space, the SBMFT Hamiltonian is then given by

$$
\hat{H}_0 = \sum_{\mathbf{k},s} \left[ \lambda_0 \left( \hat{a}_{\mathbf{k},s}^{\dagger} \hat{a}_{\mathbf{k},s} + \hat{b}_{-\mathbf{k},s} \hat{b}_{-\mathbf{k},s}^{\dagger} \right) - zQ_0 \left( \gamma_{\mathbf{k}} \hat{a}_{\mathbf{k},s}^{\dagger} \hat{b}_{-\mathbf{k},s}^{\dagger} + \gamma_{\mathbf{k}}^{*} \hat{b}_{-\mathbf{k},s} \hat{a}_{\mathbf{k},s} \right) \right],
\tag{3}
$$

up to a constant term, where $z$ is the number of nearest neighbors ($z = 4$ for the square lattice and $z = 3$ for the honeycomb lattice), where $Q_0 = \frac{J}{2} \langle \hat{A}_{\mathbf{r},\mathbf{r}+\delta} \rangle$ is the bond parameter, and where $\gamma_{\mathbf{k}} = \frac{1}{z} \sum_{\delta} e^{i\mathbf{k}\cdot\delta} = |\gamma_{\mathbf{k}}| e^{i\phi_{\mathbf{k}}}$. $\hat{H}_0$ is diagonalized by the Bogoliubov transformation

$$
\hat{a}_{\mathbf{k},s} = e^{i\phi_{\mathbf{k}}} \left( \cosh\theta_{\mathbf{k}}\, \hat{\alpha}_{\mathbf{k},s} + \sinh\theta_{\mathbf{k}}\, \hat{\beta}_{-\mathbf{k},s}^{\dagger} \right), \qquad \hat{b}_{-\mathbf{k},s} = \cosh\theta_{\mathbf{k}}\, \hat{\beta}_{-\mathbf{k},s} + \sinh\theta_{\mathbf{k}}\, \hat{\alpha}_{\mathbf{k},s}^{\dagger},
\tag{4}
$$

where $\tanh(2\theta_{\mathbf{k}}) = c_0 |\gamma_{\mathbf{k}}|$, $c_0 = zQ_0/\lambda_0$, resulting in

$$
\hat{H}_0 = \sum_{\mathbf{k},s} \omega_{\mathbf{k}} \left( \hat{\alpha}_{\mathbf{k},s}^{\dagger} \hat{\alpha}_{\mathbf{k},s} + \hat{\beta}_{-\mathbf{k},s} \hat{\beta}_{-\mathbf{k},s}^{\dagger} \right), \qquad \omega_{\mathbf{k}} = \lambda_0 \varepsilon_{\mathbf{k}}, \qquad \varepsilon_{\mathbf{k}} = \sqrt{1 - c_0^2 |\gamma_{\mathbf{k}}|^2}.
\tag{5}
$$

The dispersion relation $\omega_{\mathbf{k}}$ is plotted in Fig. 2.

The mean-field parameters $\lambda_0, Q_0$ are determined by the static self-consistency (SSC) equations, given by

$$
Q_0 = \frac{J}{2} \langle \hat{A}_{\mathbf{r},\mathbf{r}+\delta} \rangle, \qquad 2S = \sum_{s} \langle \hat{a}_{\mathbf{r},s}^{\dagger} \hat{a}_{\mathbf{r},s} \rangle = \sum_{s} \langle \hat{b}_{\mathbf{r},s}^{\dagger} \hat{b}_{\mathbf{r},s} \rangle,
\tag{6}
$$

where the expectation values are evaluated on the ground state or thermal state corresponding to $\hat{H}_0$. This can be rewritten as

$$\lambda_0 = Jz\frac{2}{N}\sum_{\mathbf{k}}\left(n_{\mathbf{k}} + \frac{1}{2}\right)\frac{|\gamma_{\mathbf{k}}|^2}{\varepsilon_{\mathbf{k}}}, \qquad S + \frac{1}{2} = \frac{2}{N}\sum_{\mathbf{k}}\left(n_{\mathbf{k}} + \frac{1}{2}\right)\frac{1}{\varepsilon_{\mathbf{k}}}, \qquad (7)$$

where $n_{\mathbf{k}} = \frac{1}{e^{\omega_{\mathbf{k}}/T}-1}$ is the Bose occupation number ($n_{\mathbf{k}} = 0$ for $T = 0$). The SSC equations are solved numerically. Using this, $\omega_{\mathbf{k}}$ has very close agreement with the LSWT result, provided the Oguchi correction is included in LSWT (See App. A). This indicates that SBMFT captures the same single-magnon frequency renormalization as the Oguchi correction.

Next we briefly state the result for the static correlation function, defined by $S(\mathbf{R}) = \frac{2}{3}\langle \hat{\mathbf{S}}_{\mathbf{r}} \cdot \hat{\mathbf{S}}_{\mathbf{r+R}}\rangle$, where we take $\mathbf{r} \in A$. Here we renormalize the correlation function by a factor $2/3$, as is commonly done in SBMFT to account for missing fluctuations from the large-$n$ expansion, and to ensure that $\langle \hat{\mathbf{S}}_{\mathbf{r}} \cdot \hat{\mathbf{S}}_{\mathbf{r+R}}\rangle = S(S+1)$ [31]. By substituting the Schwinger boson mapping Eq. (2), and reducing the 4$^{\text{th}}$ order boson expectation values to 2$^{\text{nd}}$ order by employing Wick's theorem, we obtain

$$S(\mathbf{R}) = \begin{cases} g_A^2(\mathbf{R}) - \frac{1}{4}\delta_{\mathbf{R},0}, & \text{for } \mathbf{R} \in A, \\ -g_B^2(\mathbf{R}), & \text{for } \mathbf{R} \in B, \end{cases} \qquad (8)$$

where

$$g_A(\mathbf{R}) = \frac{2}{N}\sum_{\mathbf{k}}\left(n_{\mathbf{k}} + \frac{1}{2}\right)\cos(\mathbf{k}\cdot\mathbf{R})\frac{1}{\varepsilon_{\mathbf{k}}},$$
$$g_B(\mathbf{R}) = \frac{2}{N}\sum_{\mathbf{k}}\left(n_{\mathbf{k}} + \frac{1}{2}\right)\cos(\mathbf{k}\cdot\mathbf{R} - \phi_{\mathbf{k}})\frac{|\gamma_{\mathbf{k}}|}{\varepsilon_{\mathbf{k}}}. \qquad (9)$$

## 3 Time-dependent SBMFT

Here, we study the space-time dynamics induced by a time-dependent perturbation of the exchange interaction. The light-matter Hamiltonian that is used models impulsive stimulated Raman scattering [11–14, 33] and is described as

$$\hat{H}_R(t) = \Delta J f(t)\sum_{\mathbf{r}\in A}\sum_{\boldsymbol{\delta}}(\mathbf{e}\cdot\boldsymbol{\delta})^2 \hat{\mathbf{S}}_{\mathbf{r}} \cdot \hat{\mathbf{S}}_{\mathbf{r+\boldsymbol{\delta}}}, \qquad (10)$$

where $\mathbf{e}$ the polarization vector of the light. Hence, light only perturbs exchange interactions for the bonds $\boldsymbol{\delta}$ that have a projection along $\mathbf{e}$. This type of Raman scattering Hamiltonian was originally derived based on symmetry arguments and perturbation theory for continuous laser fields [11, 12]. Recently, microscopic model calculations revealed that a similar equation holds for short laser pulses even beyond the realm of perturbative calculations [34, 35]. The above form of Raman scattering allows only for the excitation of pairs of magnons with opposite wave vectors and opposite angular momenta, such that the total momentum and spin is conserved during the light-matter interaction. Typically, the strength of the perturbation $|\Delta J| \ll J$, since below breakdown the electric field strength of the laser is (much) smaller than intrinsic electric fields of the material. Therefore we will use a linear response formulation of the SBMFT below. For impulsively stimulated Raman scattering, the function $f(t)$ mimics the time-envelope of a laser pulse (for example, a Gaussian in time), with a duration $\tau$ such that $\tau\omega_{2M} \ll 1$, $\omega_{2M}$ the maximum frequency of the two-magnon mode. In numerical simulations, this can be approximated by a square pulse $f(t) = 1$ for $0 < t \le \tau$. Below we are interested in the short-time dynamics for which the supermagnonic effect is observed. In this regime, the dynamics

induced by a single step function $f(t) = \Theta(t)$ was found to closely resemble the square pulse protocol adopted in [28]. Therefore the step function will also be used for applications of the generic time-dependent SBMFT formalism below.

For the derivation of the linear response dynamics within SBMFT, it is convenient to rewrite Eq. (10) and include it in the full time-dependent Hamiltonian as follows

$$\hat{H}(t) = J \sum_{\mathbf{r} \in A} \sum_{\delta} \big[ 1 + \eta f(t) p_{\delta} \big] \hat{\mathbf{S}}_{\mathbf{r}} \cdot \hat{\mathbf{S}}_{\mathbf{r}+\delta}, \quad \text{where} \quad p_{\delta} = 2(\mathbf{e} \cdot \delta)^2 - 1, \quad (11)$$

where $\eta \propto \Delta J / J$ is the expansion parameter. In order to derive the self-consistent linear response solution, we perform the usual Hartree-Fock decoupling of the time-dependent bond parameter $Q_{\delta}(t) = \frac{J}{2}\langle \hat{A}_{\mathbf{r},\mathbf{r}+\delta} \rangle(t)$, supplemented with the time-dependent parameter $\lambda(t)$ for both boson species. We expand the mean-field parameters as

$$\lambda(t) = \lambda_0 \big[ 1 + \eta \Delta\lambda(t) \big], \qquad Q_{\delta}(t) = Q_0 \big[ 1 + \eta \Delta Q_{\delta}(t) \big], \quad (12)$$

and subsequently expand the time-dependent Hamiltonian as $\hat{H}(t) = \hat{H}_0 + \hat{V}(t) + \mathcal{O}(\eta^2)$, where in terms of Schwinger-Bogoliubov bosons from Eq. (4)

$$\hat{V}(t) = \sum_{\mathbf{k},s} \bigg[ \delta\omega_{\mathbf{k}}(t)\big( \hat{\alpha}_{\mathbf{k},s}^{\dagger} \hat{\alpha}_{\mathbf{k},s} + \hat{\beta}_{\mathbf{k},s} \hat{\beta}_{\mathbf{k},s}^{\dagger} \big) + V_{\mathbf{k}}(t) \hat{\alpha}_{\mathbf{k},s}^{\dagger} \hat{\beta}_{-\mathbf{k},s}^{\dagger} + V_{\mathbf{k}}^{*}(t) \hat{\beta}_{-\mathbf{k},s} \hat{\alpha}_{\mathbf{k},s} \bigg], \quad (13)$$

$$\delta\omega_{\mathbf{k}}(t) = -\frac{\eta\lambda_0}{\varepsilon_{\mathbf{k}}} \bigg\{ c_0^2 |\gamma_{\mathbf{k}}| \mathrm{Re}\big[ R_{\mathbf{k}}(t) e^{-i\phi_{\mathbf{k}}} \big] - \Delta\lambda(t) \bigg\}, \quad (14)$$

$$V_{\mathbf{k}}(t) = -\eta z Q_0 \bigg\{ \frac{1}{\varepsilon_{\mathbf{k}}} \mathrm{Re}\big[ R_{\mathbf{k}}(t) e^{-i\phi_{\mathbf{k}}} \big] + i \mathrm{Im}\big[ R_{\mathbf{k}}(t) e^{-i\phi_{\mathbf{k}}} \big] - \frac{|\gamma_{\mathbf{k}}|}{\varepsilon_{\mathbf{k}}} \Delta\lambda(t) \bigg\}, \quad (15)$$

$$R_{\mathbf{k}}(t) = f(t) \Gamma_{\mathbf{k}} + \frac{1}{z} \sum_{\delta} \Delta Q_{\delta}(t) e^{i\mathbf{k}\cdot\delta}, \qquad \Gamma_{\mathbf{k}} = \frac{1}{z} \sum_{\delta} p_{\delta} e^{i\mathbf{k}\cdot\delta}. \quad (16)$$

Here, the parameters $\Delta Q(t)$ and $\Delta\lambda(t)$ still have to be determined self-consistently for each time $t$. In order to derive the self-consistency equations for these parameters, as well as for other observables of interest, the general time-dependent linear response formalism [36] will be employed. For an operator $\hat{O}$ under a time-dependent pertubation $\hat{V}(t)$, we have

$$\langle \hat{O} \rangle(t) = \langle \hat{O} \rangle_0 - i \int_{-\infty}^{t} dt' \big\langle \big[ \hat{O}_I(t), \hat{V}_I(t') \big] \big\rangle_0, \quad (17)$$

where operators in the interaction picture are given by: $\hat{A}_I(t) = e^{i\hat{H}_0 t} \hat{A}(t) e^{-i\hat{H}_0 t}$, and where $\langle \ldots \rangle_0$ denotes the expectation value with respect to the unperturbed state. Note that the $\delta\omega_{\mathbf{k}}$ term in Eq. (13) commutes with $\hat{H}_0$, hence for this perturbation in linear response we only have to consider the terms with $V_{\mathbf{k}}$. We can therefore write

$$\langle \hat{O} \rangle(t) = \langle \hat{O} \rangle_0 - i \sum_{\mathbf{k},s} \big\langle \big[ \hat{O}_I(t), \hat{\alpha}_{\mathbf{k},s}^{\dagger} \hat{\beta}_{-\mathbf{k},s}^{\dagger} \big] \big\rangle_0 \int_{-\infty}^{t} dt' V_{\mathbf{k}}(t') e^{i2\omega_{\mathbf{k}}t'}$$
$$- i \sum_{\mathbf{k},s} \big\langle \big[ \hat{O}_I(t), \hat{\beta}_{-\mathbf{k},s} \hat{\alpha}_{\mathbf{k},s} \big] \big\rangle_0 \int_{-\infty}^{t} dt' V_{\mathbf{k}}^{*}(t') e^{-i2\omega_{\mathbf{k}}t'}, \quad (18)$$

where we used the fact that $\hat{\alpha}_{\mathbf{k},s,I}(t) = e^{-i\omega_{\mathbf{k}}t} \hat{\alpha}_{\mathbf{k},s}$. Below, we first discuss how the linear response equations are used to construct the time-dependent self-consistency equations. Subsequently, we calculate the time-dependent spin correlations.

## 3.1 Time-dependent self-consistency equations

The time-dependent self-consistency (TDSC) equations determine the time evolution of the mean-field parameters in response to the perturbation. Analogously to the SSC equations Eqs. (6), they read

$$Q_{\delta}(t) = \frac{J}{2} \langle \hat{A}_{\mathbf{r},\mathbf{r}+\delta} \rangle(t), \tag{19}$$

$$2S = \sum_{s} \langle \hat{a}_{\mathbf{r},s}^{\dagger} \hat{a}_{\mathbf{r},s} \rangle(t) = \sum_{s} \langle \hat{b}_{\mathbf{r}',s}^{\dagger} \hat{b}_{\mathbf{r}',s} \rangle(t). \tag{20}$$

As the operators $\hat{a}_{\mathbf{r},s}^{\dagger} \hat{a}_{\mathbf{r},s}$ and $\hat{b}_{\mathbf{r}',s}^{\dagger} \hat{b}_{\mathbf{r}',s}$ commute with the spin operators, they also commute with the full Hamiltonian Eq. 11. Hence, $\langle \hat{a}_{\mathbf{r},s}^{\dagger} \hat{a}_{\mathbf{r},s} \rangle, \langle \hat{b}_{\mathbf{r}',s}^{\dagger} \hat{b}_{\mathbf{r}',s} \rangle$ are constant in time, and therefore Eq. (20) reduces to the SSC equation. Consequently, $\Delta\lambda(t)$ is always zero, and we only need to consider Eq. (19). In linear response, the time-dependent expectation value is evaluated using Eq. (18). The $\mathcal{O}(\eta^{0})$ result yields the SSC equation, and the $\mathcal{O}(\eta)$ result determines the time-dependent first-order correction $\Delta Q_{\delta}(t)$.

The TDSC equations are a set of integral equations, and cannot be solved analytically in the time domain. However, an analytical solution exists in the Laplace domain, which is derived in App.(C). Below we use the notation that for a function $g(t)$ in real time we have the Laplace transform $\tilde{g}(s)$, in the conjugate variable $s$. In particular, we write $\Delta\tilde{Q}_{\delta}(s) = p_{\delta}\tilde{f}(s)\tilde{q}(s)$. To convert back to the time domain, we numerically calculate the inverse using the algorithm of Ref. [37], which is based on a Fourier series method. To speed up the series convergence, this algorithm is supplemented with the epsilon algorithm [38].

The dynamics of observables and other quantities can be expressed in terms of $q_{\mathrm{r}}(t) = \mathrm{Re}\, q(t)$ and $q_{\mathrm{i}}(t) = \mathrm{Im}\, q(t)$. Fig. 3 shows these functions in the time and Laplace domain, for both the square and honeycomb lattice. We work in dimensionless time and frequency units, scaled by the characteristic frequency $\omega_{2\mathrm{M}} = 2\lambda_{0}$. For both lattices, $q(t)$ is a damped oscillatory function with a frequency corresponding to the two-magnon excitation. The damping is much stronger in the honeycomb case, which we ascribe to stronger magnon-magnon interactions. As we will see in the next section, this significantly affects the behavior of the spin correlation function at small distances. For increasing $S$, the two-magnon frequency shifts towards $\omega_{2\mathrm{M}} = 2\lambda_{0}$, and the amplitude decreases. This is to be expected, as the $S \rightarrow \infty$ limit should resemble LSWT, where no mean-field dynamics is present.

The physics of the time-dependent SBMFT solution can be understood as follows. In the absence of dynamical corrections to $\Delta Q(t)$, there is no qualitative difference with the results from LSWT. The dynamical correction of the bond parameter therefore involve additional effects of magnon-magnon interactions that are not present in LSWT. Hence, by taking into account magnon-magnon interactions, the excited state spectrum of magnon-pairs changes and is qualitatively distinct from the sum of two single-magnon spectra. The results in Fig. 3 shows that the dynamical contribution $\Delta Q(t)$ is heavily damped. Hence the dynamical contribution to the two-magnon spectrum involves a quasi-bound state. The emergence of this quasi-bound state is key to understanding the correlation dynamics and is further discussed in the next section.

## 3.2 Correlation function dynamics

Next we proceed with the calculation of the time-dependent spin correlation function, defined by

$$C(\mathbf{R}, t) = \left[ \langle \hat{\mathbf{S}}_{\mathbf{r}} \cdot \hat{\mathbf{S}}_{\mathbf{r}+\mathbf{R}} \rangle(t) - \langle \hat{\mathbf{S}}_{\mathbf{r}} \rangle(t) \cdot \langle \hat{\mathbf{S}}_{\mathbf{r}+\mathbf{R}} \rangle(t) \right] - \left[ \langle \hat{\mathbf{S}}_{\mathbf{r}} \cdot \hat{\mathbf{S}}_{\mathbf{r}+\mathbf{R}} \rangle_{0} - \langle \hat{\mathbf{S}}_{\mathbf{r}} \rangle_{0} \cdot \langle \hat{\mathbf{S}}_{\mathbf{r}+\mathbf{R}} \rangle_{0} \right], \tag{21}$$

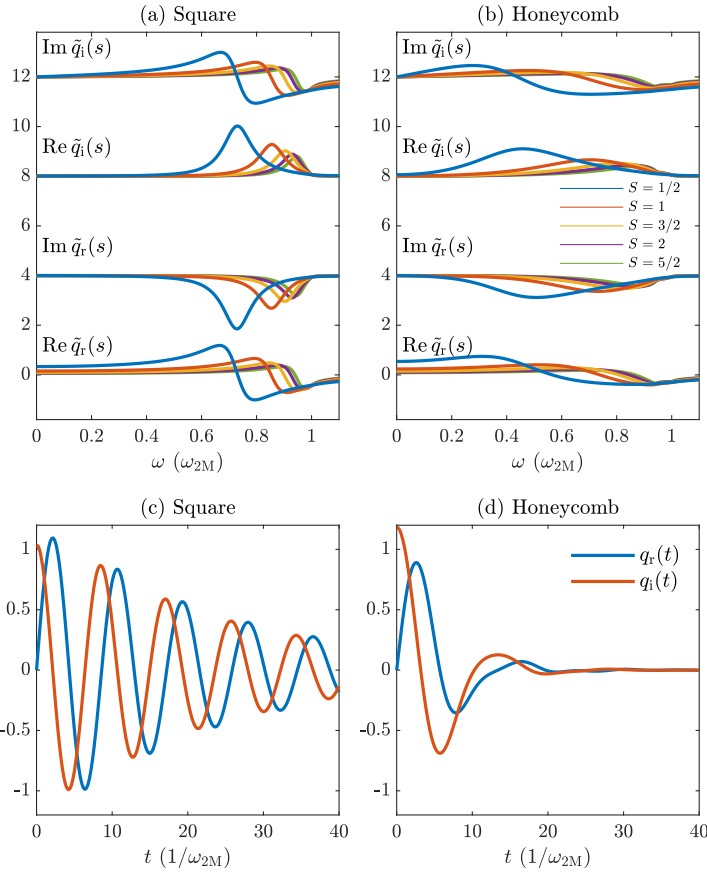

Figure 3: (a)/(b): Spectral representation of $\tilde{q}(s)$ for the square and honeycomb lattice, for a range of spin values $S$, for $L = 200$. The spectrum is evaluated along $s = 0.1J + i\omega$. (c)/(d) Real-time representation of $q(t)$ for the square and honeycomb lattice for $S = 1/2$, $L = 60$. Here we denote $q_r(t) = \text{Re}\, q(t)$ and $q_i(t) = \text{Im}\, q(t)$. Units are scaled with the 2-magnon frequency $\omega_{2M} = 2\lambda_0$.

where we take $\mathbf{r} \in A$. In SBMFT, the single-spin expectation values are zero due to rotational symmetry. Furthermore, we again renormalize by a factor $2/3$ similarly to the static case in Eq. (8). This can done by omitting the $S^z S^z$ term, because the $x, y, z$ terms are equal due to rotational symmetry. Therefore we can write

$$C(\mathbf{R}, t) = \langle \hat{O} \rangle(t) - \langle \hat{O} \rangle_0, \quad \text{where} \quad \hat{O} = \frac{1}{2} \frac{2}{N} \sum_{\mathbf{r} \in A} \left[ \hat{S}_{\mathbf{r}}^+ \hat{S}_{\mathbf{r+R}}^- + \hat{S}_{\mathbf{r}}^- \hat{S}_{\mathbf{r+R}}^+ \right]. \tag{22}$$

This quantity is calculated in linear response using Eq. (18). After substituting the Schwinger boson mapping Eq. (2) into Eq. (22), and subsequently reducing the resulting 6$^{th}$ order boson expectation values to 2$^{nd}$ order by employing Wick's theorem, the final result is

$$C(\mathbf{R}, t) = \begin{cases} -g_A(\mathbf{R}) u_A(\mathbf{R}, t), & \text{for } \mathbf{R} \in A, \\ g_B(\mathbf{R}) u_B(\mathbf{R}, t), & \text{for } \mathbf{R} \in B, \end{cases} \tag{23}$$

where $g_{A/B}(\mathbf{R}, t)$ is given by Eq. (9), and where

$$u_A(\mathbf{R}, t) = \frac{2}{N} \sum_{\mathbf{k}} \left( n_{\mathbf{k}} + \frac{1}{2} \right) \frac{c_0 |\gamma_{\mathbf{k}}|}{\varepsilon_{\mathbf{k}}} \cos(\mathbf{k} \cdot \mathbf{R}) \operatorname{Im} X_{\mathbf{k}}(t),$$

$$u_B(\mathbf{R}, t) = \frac{2}{N} \sum_{\mathbf{k}} \left( n_{\mathbf{k}} + \frac{1}{2} \right) \left[ \frac{1}{\varepsilon_{\mathbf{k}}} \cos(\mathbf{k} \cdot \mathbf{R} - \phi_{\mathbf{k}}) \operatorname{Im} X_{\mathbf{k}}(t) - \sin(\mathbf{k} \cdot \mathbf{R} - \phi_{\mathbf{k}}) \operatorname{Re} X_{\mathbf{k}}(t) \right], \quad (24)$$

$$X_{\mathbf{k}}(t) = -4 \int_{-\infty}^{t} dt' V_{\mathbf{k}}(t') e^{-i2\omega_{\mathbf{k}}(t-t')}.$$

The dynamics of the correlation function is partly due to the mean-field dynamics, as $V_{\mathbf{k}}(t)$ contains the dynamics of the parameter $q(t)$ in Eq. (C.8). All time dependence is contained in the factor $X_{\mathbf{k}}(t)$. Similarly to the time-dependent mean-field parameters, we evaluate this in Laplace space. Using Eq. (C.9) of App. C the following expression is obtained

$$\tilde{X}_{\mathbf{k}}(s) = 2\eta c_0 \tilde{f}(s) \frac{\bar{s} - i\varepsilon_{\mathbf{k}}}{\varepsilon_{\mathbf{k}}^2 + \bar{s}^2} \left( \frac{1}{\varepsilon_{\mathbf{k}}} \operatorname{Re} \left\{ \left[ 1 + \tilde{q}(s) \right] \Gamma_{\mathbf{k}} e^{-i\phi_{\mathbf{k}}} \right\} + i \operatorname{Im} \left\{ \left[ 1 + \tilde{q}(s) \right] \Gamma_{\mathbf{k}} e^{-i\phi_{\mathbf{k}}} \right\} \right). \quad (25)$$

Some terms in the above expression vanish after summing over $\mathbf{k}$ in Eq. (24) due to antisymmetry of the summand. The only terms that survive are

$$\operatorname{Im} \tilde{X}_{\mathbf{k}}(s) : -2\eta c_0 \frac{\tilde{f}(s)}{\varepsilon_{\mathbf{k}}^2 + \bar{s}^2} \left[ 1 + \tilde{q}_r(s) - \bar{s} \tilde{q}_i(s) \right] \operatorname{Re} \left( \Gamma_{\mathbf{k}} e^{-i\phi_{\mathbf{k}}} \right),$$

$$\operatorname{Re} \tilde{X}_{\mathbf{k}}(s) : \quad 2\eta c_0 \frac{\tilde{f}(s)}{\varepsilon_{\mathbf{k}}^2 + \bar{s}^2} \left\{ \varepsilon_{\mathbf{k}} \left[ 1 + \tilde{q}_r(s) \right] - \frac{\bar{s}}{\varepsilon_{\mathbf{k}}} \tilde{q}_i(s) \right\} \operatorname{Im} \left( \Gamma_{\mathbf{k}} e^{-i\phi_{\mathbf{k}}} \right). \quad (26)$$

Note that for the square lattice, $\operatorname{Im}(\Gamma_{\mathbf{k}} e^{-i\phi_{\mathbf{k}}}) = 0$.

Next we will numerically evaluate the correlation function both in real space and time: $C(\mathbf{R}, t)$ and in reciprocal space and Laplace space: $\tilde{C}(\mathbf{q}, s)$, and analyze the results. The inverse Laplace transform is calculated in the same way as before. Up to now, the derivations have been done for arbitrary pulse profile $f(t)$. In the remainder of this paper we will restrict ourselves to a quench-like perturbation, where the pulse profile is a step function: $f(t) = \Theta(t)$. We also restrict ourselves to the polarization $\mathbf{e} = (1, 0)$. Further analysis of dependencies on e.g. polarization, pulse profile, and temperature is not addressed here. To understand the effect of magnon-magnon interactions, the SBMFT results will also be compared to LSWT (see App. A) in which interactions between magnons are neglected (apart from the Oguchi correction).

Firstly, we analyze the spatial spreading of the correlation function after the perturbation. Figs. 4,5 show snapshots in time of the correlation function. While at small $\mathbf{R}$ the square lattice spatial pattern is exactly antisymmetric along $x = \pm y$ (in linear response, as proven in [26]), the honeycomb lattice spatial pattern is highly asymmetric. This difference also plays a role in the supermagnonic effect, as we will see in the next section. At large $\mathbf{R}$, we observe an approximately circular wavefront for both lattices, with $C(\mathbf{R}, t) \sim \cos(2\phi_{\mathbf{R}})\cos(2\phi_{\mathbf{e}})$ for the square lattice, and $C(\mathbf{R}, t) \sim \cos[2(\phi_{\mathbf{R}} - \phi_{\mathbf{e}})]$ for the honeycomb lattice. Further analysis of the spatial symmetry of the square lattice correlation function can be found in Ref. [26]. The large-$\mathbf{R}$ results are only shown for LSWT, as the SBMFT calculation is computationally too expensive. However, we observed that already at smaller $\mathbf{R}$ (but larger than in Fig. 4), LSWT and SBMFT show qualitatively the same spatial pattern, and we expect the same to hold for larger $\mathbf{R}$.

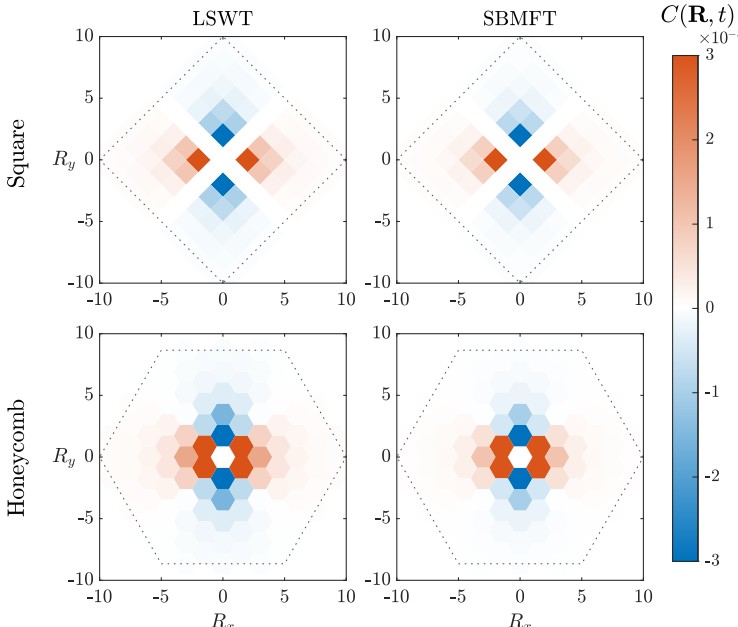

Figure 4: Spatial profile at $t = 3\,(1/\omega_{2M})$ of the small-**R** spin correlations, for the square and honeycomb lattice, as calculated with LSWT and SBMFT for a system size $N = 2L^2$, $L = 20$, showing only $\mathbf{R} \in A$. The square lattice result is perfectly antisymmetric along $x = \pm y$, while the honeycomb lattice result is highly asymmetrical at short distances. To improve readability, the colormap is clipped at $3 \cdot 10^{-3}$.

Fig. 6 shows the time evolution of the nearest neighbor correlations for $\mathbf{R} = (-1, 0)$ for SBMFT (with spin dependence) and LSWT. The $y$-axis is scaled by $1/S$ as the LSWT result is proportional to $S$ (See App. A). This figure reveals a large quantitative difference between $S = 1/2$ SBMFT and LSWT, the former having greater amplitude and larger damping. Moreover, it clearly shows a convergence towards the LSWT result in the limit $S \to \infty$. This result demonstrates the effect of magnon-magnon interactions on the nearest neighbor correlations. As was shown in Ref. [26], the interactions influence magnon propagation at small distances, resulting in the supermagnonic effect. Fig. 6 also reveals that in the honeycomb lattice, the nearest neighbor correlations are even more significantly influenced by magnon-magnon interactions than in the square lattice, resulting in a larger lowering of the frequency, and a much greater damping. This mirrors the result of Fig. 3. This higher damping also influences the supermagnonic effect, as we will see in the next section.

Fig. 7 shows the time evolution of the $S = 1/2$ correlation function beyond nearest neighbor, for SBMFT only. We consider only correlations along the $x$-axis and $y$-axis. For the square lattice we only show the result for the $x$-axis because of the exact antisymmetry between $x$ and $y$. For the honeycomb lattice there is a significant anisotropy between the $x$ and $y$ correlations, which relates to the asymmetry seen in Fig. 4 and the absence of an exact antisymmetry. Furthermore, Fig. 7 shows a clear increase in the time until the correlations have reached their maximum for increasing distance. For the square lattice, the correlations show a period of 2 in the sense that the after 2 unit lengths, the correlation function returns to approximately the same waveform, but shifted in time. In the same sense, the honeycomb ($x$) correlations show a period of 3, and the honeycomb ($y$) correlations a period of 1. While for the square lattice the waveforms within a single period are approximately equal (apart from a different amplitude and sign), for the honeycomb ($x$) result, the waveforms within a single period are qualitatively very different, which ultimately stems from the broken inversion symmetry of the lattice.

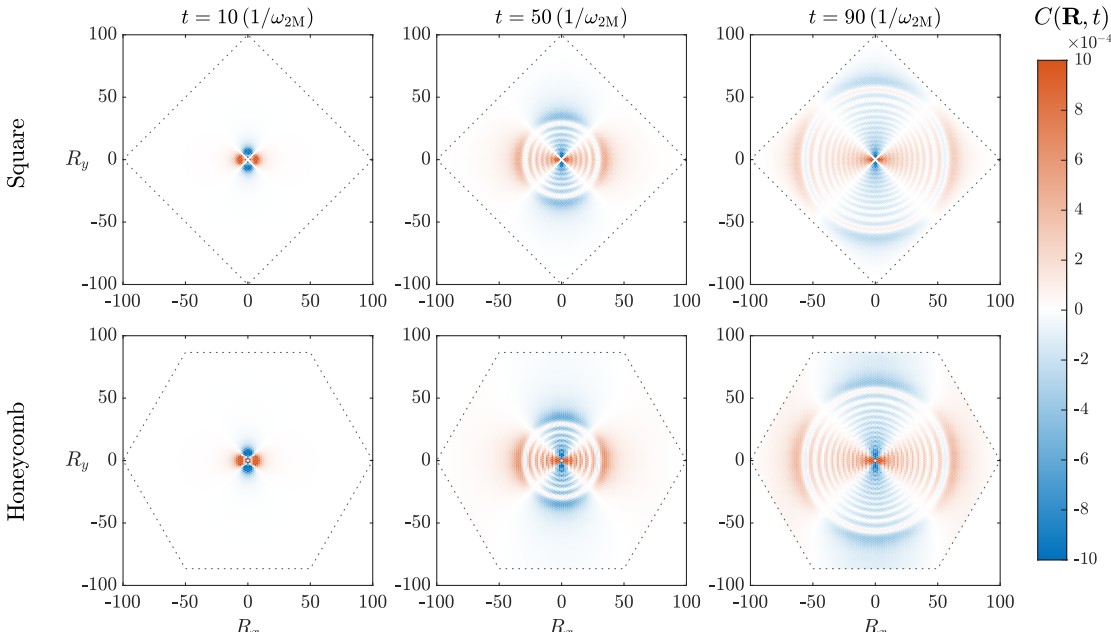

Figure 5: Consecutive snapshots in time of the large-$\mathbf{R}$ postquench dynamics of spin correlations for the square and honeycomb lattice, for $S = 1/2$, as calculated with LSWT for a system size $N = 2L^2$, $L = 100$, showing only $\mathbf{R} \in A$. At large distances, the wavefront is approximately circular, with sinusoidal radial dependence. To improve readability, the colormap is clipped at $10^{-3}$.

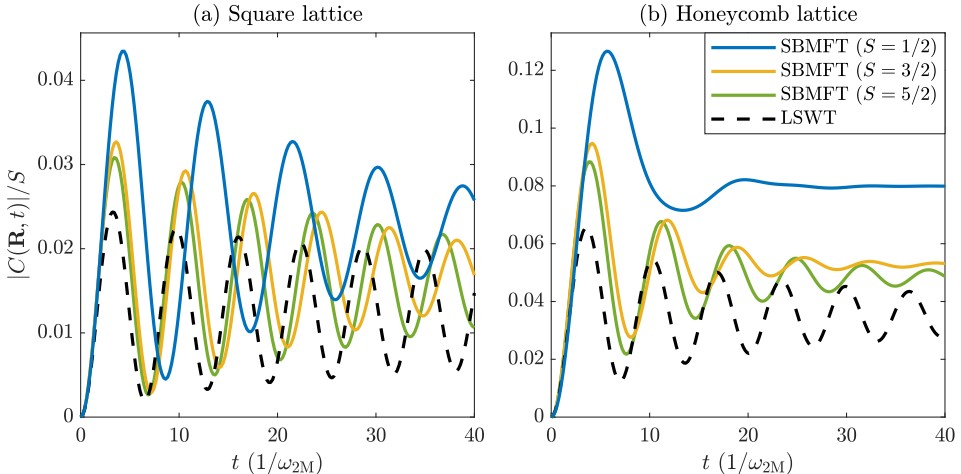

Figure 6: Nearest neighbour correlations for $\mathbf{R} = (-1, 0)$ for (a) the square and (b) the honeycomb lattice, for a system size $N = 2L^2$, $L = 40$. The SBMFT result converges to LSWT as $S \to \infty$, and the honeycomb result shows much larger damping than the square result.

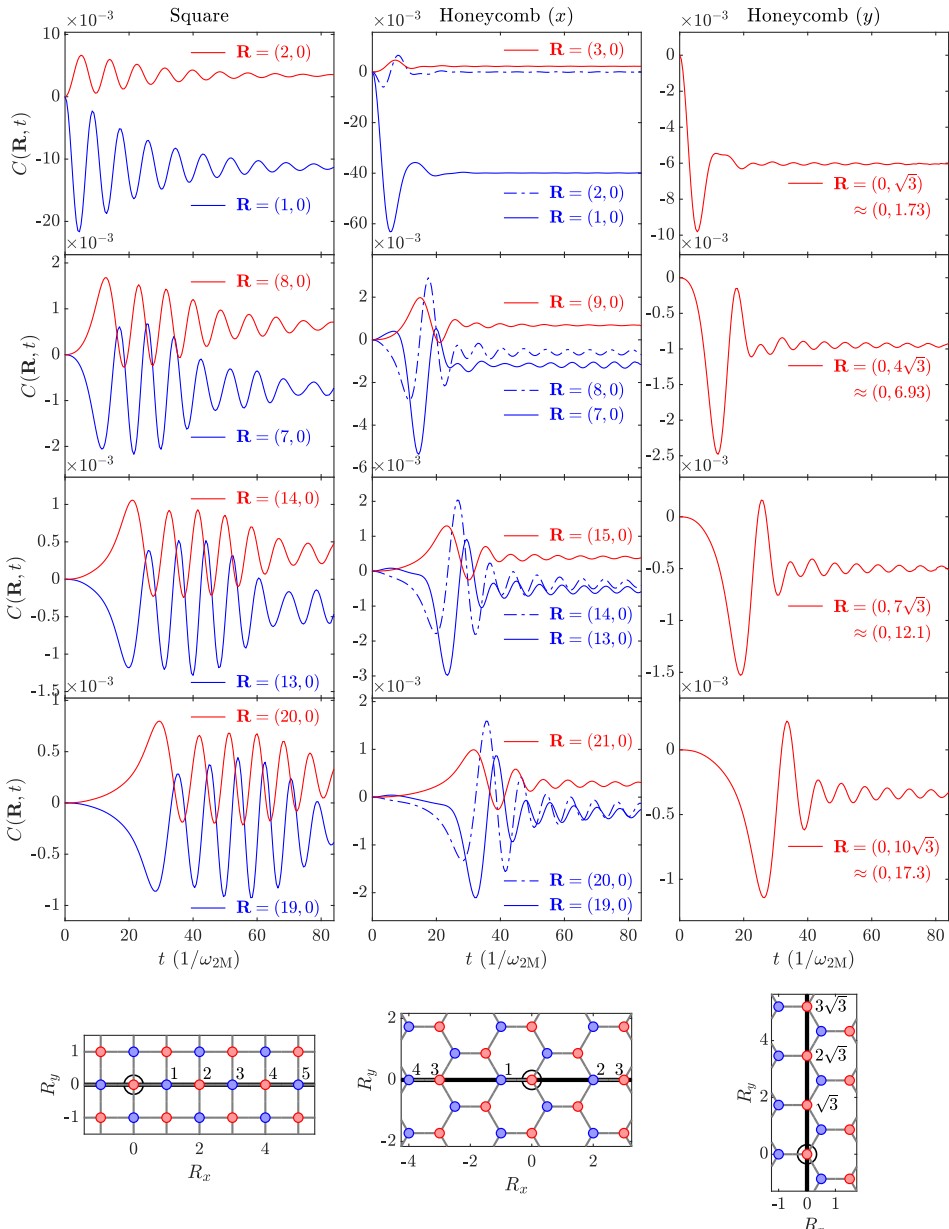

Figure 7: Time evolution of the $S = 1/2$ correlation function for spins separated by various $\mathbf{R}$ along a single axis, for the square lattice along the $x$-axis, and for the honeycomb lattice along both the $x$-axis and $y$-axis, for a system size $N = 2L^2$, $L = 100$. The bottom subfigures show the spin lattices, where the origin is black encircled, and with the first few spins along a single axis labeled with their absolute distances to the origin.

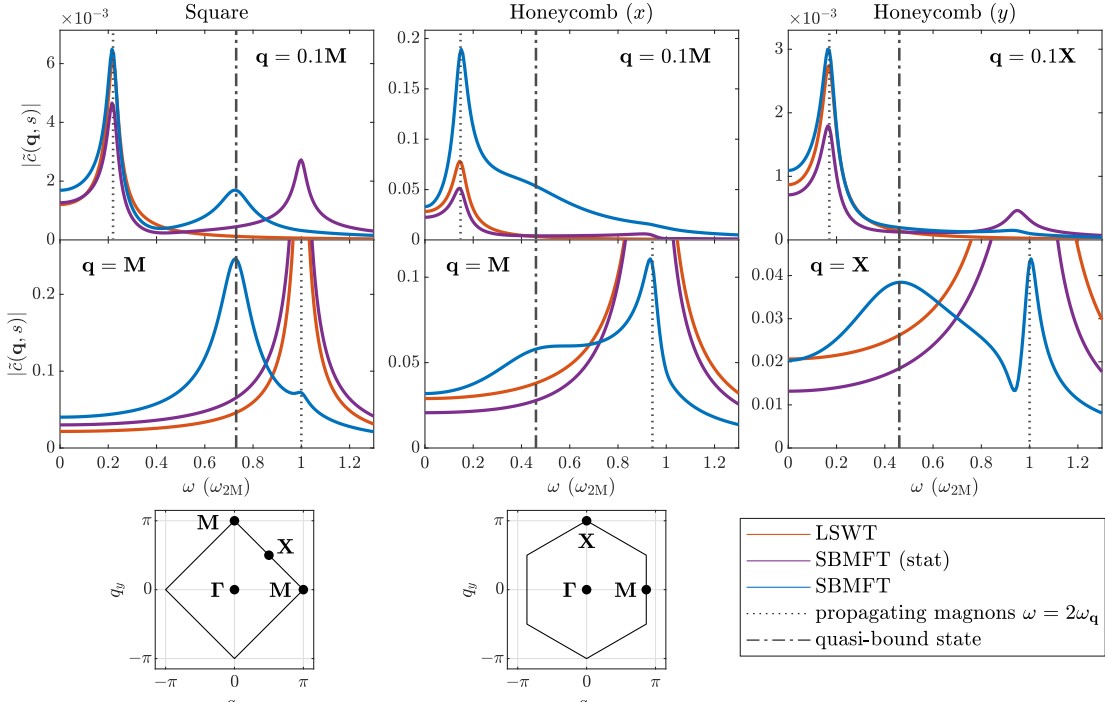

Figure 8: Spectral representation of the correlation function for the square lattice (along $x$) and honeycomb lattice (along $x$ and $y$), for a system size $N = 2L^2$, $L = 120$. The spectrum is evaluated along $s = 0.1J + \mathrm{i}\omega$. The bottom figures show the Brillouin zones and their high-symmetry points.

To gain more insight into the constituent excitations of the correlation function, we study its spectral representation. In Laplace space, the correlation function factorizes as $\tilde{C}(s) = \tilde{f}(s)\tilde{c}(s)$, where $f$ is the pulse profile. Fig. 8 shows the function $\tilde{c}(\mathbf{q}, s)$ in reciprocal and Laplace space, again along the $x$-axis for the square lattice (the antisymmetry also holds in reciprocal space) and along both the $x$-axis and the $y$-axis for the honeycomb lattice. Results are shown for both LSWT and SBMFT, as well as the the 'static' SBMFT, which corresponds to replacing the time-dependent $q_\mathrm{r}(t)$ and $q_\mathrm{i}(t)$ by the constant values which they attain in the limit $t \to \infty$ for a step function pulse profile (see App. (B)). We claim that SBMFT (static) corresponds to the SBMFT without magnon-magnon interactions beyond the bare magnon renormalization (similar to the Oguchi correction in LSWT), and that the function $q(t)$ captures all dynamical magnon-magnon interactions. We observe that all results show a peak at the twice the single magnon frequency, corresponding to an excitation of two non-interacting propagating magnons. Additionally, the SBMFT results show a peak at a fixed two-magnon frequency (independent of $\mathbf{q}$, similar to Fig. 3). This peak describes a distinct quasi-bound state of local spin-flip excitations caused by magnon-magnon interactions. A more detailed analysis of the square lattice result can be found in Ref. [26]. Finally, we observe that at small wavelengths, the propagating magnons are always dominant, while at high wavelengths, the quasi-bound state is dominant for square and honeycomb ($y$), but not for honeycomb ($x$).

## 4 Supermagnonic propagation

In this section, we analyze the magnon propagation speed, as given by the slope of the light cone. It was shown in Ref. [26] that for the square lattice, the velocity at small distances is

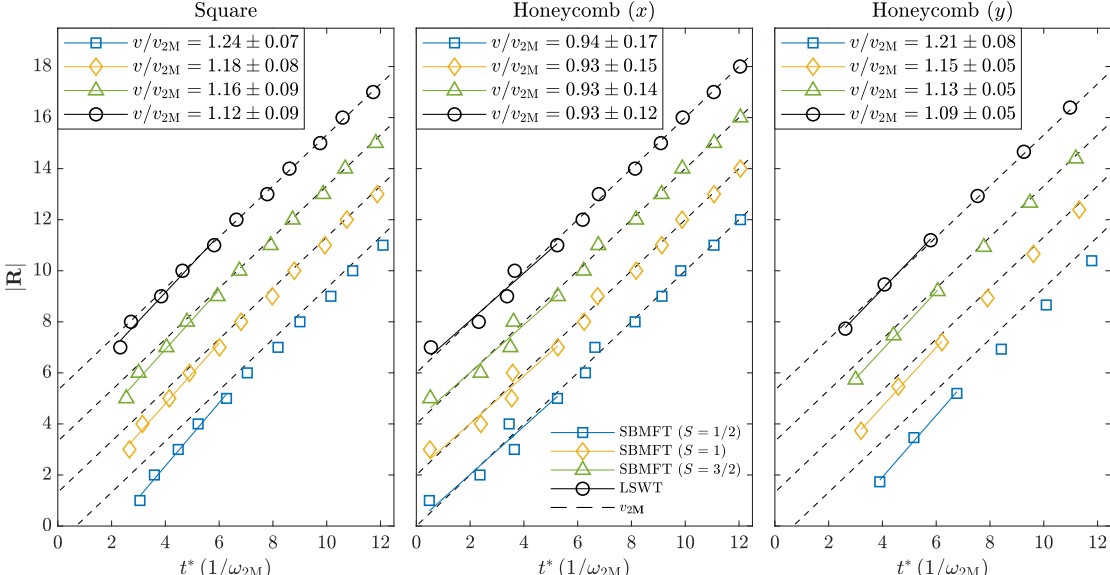

Figure 9: Supermagnonic fit showing the arrival times $t^*$ of the first extrema of the correlation function for SBMFT (several values of spin $S$) and LSWT. The black dashed line indicates twice the highest magnon group velocity $v_{2M}$. The system size is $N = 2L^2$, $L = 30$. The honeycomb ($x$) results are rescaled by a large-$\mathbf{R}$ correction. To improve readability, the data have been shifted vertically by multiples of $\mathbf{R} = 2$.

higher than what can be expected from the single-magnon group velocity, which was dubbed the *supermagnonic effect*. Moreover, this was shown to be the result of extraordinarily strong magnon-magnon interactions. This result was obtained using Neural Quantum States, SBMFT, and LSWT.

Here, we investigate the supermagnonic effect in the honeycomb lattice using SBMFT and LSWT, and relate the findings to the spectral representation of the correlation function. The propagation speed is extracted by fitting the distance $\mathbf{R}$ against the arrival time $t^*$, defined by the time at which the first extremum of the correlation function $C(\mathbf{R}, t)$ appears. This is done for distances up to $|\mathbf{R}| \approx 5$. Fig (9) shows the arrival times and corresponding fits for the square lattice, and the honeycomb lattice along the $x$ and $y$ axis. The square lattice result is equivalent for $x$ and $y$ due to reflection antisymmetry along $x = \pm y$ of the correlation function. For the square lattice, the slope exhibits a bending when going from small to larger distances ($\mathbf{R} > 5$), as was already shown in Ref. [26]. For $|\mathbf{R}| \leq 5$, the fitted velocity is $24 \pm 7\%$ higher than $v_{2M}$ in the case with strongest magnon-magnon interactions ($S = 1/2$). The explanation for this effect is that the small-$\mathbf{R}$ correlations are dominated by the large-$\mathbf{q}$ correlations, at which the quasi-bound state (QBS) is dominant (see Fig. (8)). This QBS excitation has a lower frequency due to magnon-magnon interactions, thereby delaying the arrival time of the first extremum of $C(\mathbf{R}, t)$. Larger $\mathbf{R}$ correspond to smaller $\mathbf{q}$, where the QBS is suppressed, but (non-interacting) propagating magnon (PM) modes are dominant. We say that there is a large contrast between small and large $\mathbf{q}$.

The data points for honeycomb ($x$) have been have been subjected to a correction. In Fig. (7), we observe that for honeycomb ($x$), the positions of the first extrema are not increasing monotonically as a function of $|\mathbf{R}|$. Instead, it shows a period of 3, after which the wave profiles are approximately the same as in the previous period, but delayed in time. This is a result of the broken inversion symmetry. Note that this effect is also present in LSWT and is thus independent of magnon-magnon interactions. In order to obtain data points that allow for a proper fit, we correct the small-$\mathbf{R}$ values by subtracting the large-$\mathbf{R}$ trend. At large $\mathbf{R}$, we

expect no effect of the interactions, but the characteristic period is still present and constant. The correction amounts to $t^*(\mathbf{R}) \to t^*(\mathbf{R}) - \left(t^*(\mathbf{R_l}) - \frac{\mathbf{R_l}}{v_{2M}}\right)$, where $\mathbf{R_l}$ is large and a multiple of 3 away from $\mathbf{R}$. For $|\mathbf{R}| = 1, 4, 7, 10$, we take $\mathbf{R_l} = 13$, for $|\mathbf{R}| = 2, 5, 8, 11$, we take $\mathbf{R_l} = 14$, and for $|\mathbf{R}| = 3, 6, 9, 12$, we take $\mathbf{R_l} = 15$. These values are large enough for the effects of magnon-magnon interactions to have vanished. For honeycomb ($y$), all data points are for spins on the same sublattice, so no correction is needed.

After the correction, the honeycomb lattice along $x$ shows no discernible supermagnonic effect for $|\mathbf{R}| \leq 5$. The SBMFT results for all spin values are qualitatively the same as LSWT, all of which have a slope approximately equal to $v_{2M}$. The honeycomb lattice along $y$ however does show a bending of the slope. Thus there is a supermagnonic effect, and the velocity fitted over $|\mathbf{R}| \leq$ is $21 \pm 8\%$ higher than $v_{2M}$ in the case with strongest magnon-magnon interactions ($S = 1/2$). Similar to the square lattice, we observe a decrease of the supermagnonic effect as $S$ increases. Thus the supermagnonic effect in the honeycomb lattice is highly anisotropic. These findings can again be related to the spectral representation of the correlation function. The small-$\mathbf{R}$ correlations along $x$ are dominated by the high-$\mathbf{q}$ correlations along $x$, and the small-$\mathbf{R}$ correlations along $y$ are dominated by the high-$\mathbf{q}$ correlations along $y$. From Fig. (8), we see that along $y$ the QBS dominates at high $\mathbf{q}$ and the PMs dominate at small $\mathbf{q}$ (high contrast), while along $x$ the QBS and PMs are approximately equally abundant for both large and small $\mathbf{q}$ (low contrast). Note that along $x$, even though the amplitude of the QBS peak is lower than the PM peak, its integrated area is comparable due to the higher peak width. From these results we find that the amplitude of the supermagnonic effect does not scale directly with the amount of quantum fluctuations or the strength of magnon-magnon interactions, as one might expect, but rather with the amount of contrast between the QBS and PMs at small and large $\mathbf{q}$, which is not only dependent on the amount of interactions, but also on the lattice geometry. This is furthermore confirmed by the observation that for increasing $S$, the QBS increases in frequency and is gradually replaced by PM modes, resulting in a lower contrast. Finally we note that the 'static' SBMFT results of $t^*$ (not explicitly shown here) coincide almost exactly with the LSWT results, confirming that the supermagnonic effect is a result of magnon-magnon interactions.

## 5 Conclusion

In summary, we developed a methodology for calculating the space-time dynamics of spin correlations within the time-dependent Schwinger-boson mean-field theory (SBMFT) in the linear-response regime. This methodology allows us to compare quantum effects originating from both the discrete lattice and the discrete nature of the quantum spins themselves within one framework. As application, we studied the propagation of spin correlations and the supermagnonic effect, in which spin correlations transiently propagate faster than the highest magnon group and phase velocity. We confirm that the propagation pattern is qualitatively the same in SBMFT and linear spin-wave theory (LSWT), featuring a circular wavefront at large length scale irrespective of the underlying lattice. To the contrary, the propagation velocity at short length and time scales differs significantly. Systematic comparison between the square lattice and honeycomb lattice and variation of the quantum spin number S allowed us to gain a deeper understanding on the origin of the supermagnonic propagation. We showed that within SBMFT, the supermagnonic effect arises from the interplay between the propagating pairs of magnons that are also present in linear spin-wave theory and an additional quasi-bound state arising from magnon-magnon interactions. We found that the latter is strongly dependent on the quantum nature of the spins and on the coordination and geometry of the underlying lattice. As a result, the strength of the supermagnonic effect is not simply propor-

tional to strength of the quantum fluctuations, but rather relies on the presence of distinct regimes of dominance. The square lattice features clear dominance of the quasi-bound state at high momentum $q$ with only minor propagating modes, while for small momentum $q$ the situation is reversed. Conversely, in the honeycomb lattice, the overall strength of the quantum fluctuations is enhanced, but even at high momentum both the quasi-bound state and the propagating magnons remain visible. In this case, the supermagnonic effect is weaker and even almost completely absent for the correlations spreading along the horizontal direction. It will be very interesting to confirm this interpretation with more accurate simulations of the spin dynamics in the honeycomb lattice, for example using neural neural network quantum states [26]. Furthermore, tuning the contrast between quasi-bound states and propagating magnons suggest a prospective strategy for further enhance the supermagnonic effect. This may be feasible by investigating magnon-magnon interactions for non-linear dynamics [39] and by extending the Heisenberg model to include four-spin interactions. The latter have been studied before using quantum Monte Carlo simulations [40] and also have importance in van der Waals magnets [41]. Moreover, such studies may accelerate experimental verification. For example, when the supermagnonic regime is enhanced and extends to larger length and time scales it may reduce the required spatio-temporal resolution for experimental detection, providing fascinating perspectives to challenge the fundamental speed limits of magnon propagation.

## Acknowledgments

Discussions with G. Fabiani are greatly acknowledged.

**Funding information**    This is part of the Shell-NWO/FOM-initiative "Computational sciences for energy research" of Shell and Chemical Sciences, Earth and Life Sciences, Physical Sciences, FOM and STW, and received funding from the European Research Council under ERC Grant Agreement No. 856538 (3D-MAGiC), and Horizon Europe project No. 101070290 (NIMFEIA).

## A   LSWT

In this section, the results of the main text concerning linear spin wave theory and the corresponding dynamics of spin correlations are derived. We consider the low-order Holstein-Primakoff transformation

$$
\begin{array}{ll}
\text{for } \mathbf{r} \in A\,, & \text{for } \mathbf{r} \in B\,, \\
\hat{S}_{\mathbf{r}}^{+} = \sqrt{2S}\,\hat{a}_{\mathbf{r}}\,, & \hat{S}_{\mathbf{r}}^{+} = -\sqrt{2S}\,\hat{b}_{\mathbf{r}}^{\dagger}\,, \\
\hat{S}_{\mathbf{r}}^{-} = \sqrt{2S}\,\hat{a}_{\mathbf{r}}^{\dagger}\,, & \hat{S}_{\mathbf{r}}^{-} = -\sqrt{2S}\,\hat{b}_{\mathbf{r}}\,, \\
\hat{S}_{\mathbf{r}}^{z} = S - \hat{a}_{\mathbf{r}}^{\dagger}\hat{a}_{\mathbf{r}}\,, & \hat{S}_{\mathbf{r}}^{z} = -S + \hat{b}_{\mathbf{r}}^{\dagger}\hat{b}_{\mathbf{r}}\,.
\end{array}
\tag{A.1}
$$

Using this transformation, the unperturbed Hamiltonian in momentum space is (up to a constant term):

$$
\hat{H}_0 = J \sum_{\mathbf{r} \in A} \sum_{\boldsymbol{\delta}} \hat{\mathbf{S}}_{\mathbf{r}} \cdot \hat{\mathbf{S}}_{\mathbf{r}+\boldsymbol{\delta}} = J \sum_{\mathbf{k}} \left( \hat{a}_{\mathbf{k}}^{\dagger}\hat{a}_{\mathbf{k}} + \hat{b}_{-\mathbf{k}}\hat{b}_{-\mathbf{k}}^{\dagger} - \gamma_{\mathbf{k}}\hat{a}_{\mathbf{k}}^{\dagger}\hat{b}_{-\mathbf{k}}^{\dagger} - \gamma_{\mathbf{k}}^{*}\hat{b}_{-\mathbf{k}}\hat{a}_{\mathbf{k}} \right).
\tag{A.2}
$$

We diagonalize $\hat{H}_0$ with the Bogoliubov transformation

$$
\hat{a}_{\mathbf{k}} = e^{i\phi_{\mathbf{k}}}\left( c_{\mathbf{k}}\hat{\alpha}_{\mathbf{k}} + s_{\mathbf{k}}\hat{\beta}_{-\mathbf{k}}^{\dagger} \right), \qquad \hat{b}_{-\mathbf{k}} = c_{\mathbf{k}}\hat{\beta}_{-\mathbf{k}} + s_{\mathbf{k}}\hat{\alpha}_{\mathbf{k}}^{\dagger},
\tag{A.3}
$$

where $c_{\mathbf{k}} = \cosh\theta_{\mathbf{k}}, s_{\mathbf{k}} = \sinh\theta_{\mathbf{k}}, \tanh(2\theta_{\mathbf{k}}) = |\gamma_{\mathbf{k}}|$. The result is

$$\hat{H} = \sum_{\mathbf{k}} \omega_{\mathbf{k}}\left(\hat{\alpha}_{\mathbf{k}}^{\dagger}\hat{\alpha}_{\mathbf{k}} + \hat{\beta}_{-\mathbf{k}}\hat{\beta}_{-\mathbf{k}}^{\dagger}\right), \qquad \omega_{\mathbf{k}} = JSz\varepsilon_{\mathbf{k}}, \qquad \varepsilon_{\mathbf{k}} = \sqrt{1 - |\gamma_{\mathbf{k}}|^2}. \tag{A.4}$$

We include the Oguchi correction to the single-magnon spectrum, which is given by the renormalization $\omega_{\mathbf{k}} \to Z\omega_{\mathbf{k}}$, where

$$Z = 1 + \frac{1}{2S}\left(1 - \frac{2}{N}\sum_{\mathbf{k}}\varepsilon_{\mathbf{k}}\right) = \begin{cases} 1.158, & \text{square lattice,} \\ 1.210, & \text{honeycomb lattice.} \end{cases} \tag{A.5}$$

This captures the most simple effect of magnon-magnon interactions. Around $\mathbf{k} = 0$, the dispersion can be written as $\omega_{\mathbf{k}} \approx \frac{JSZz}{\sqrt{2}}|\mathbf{k}|$. Thefore, the magnon group velocity at $\mathbf{k} = 0$ is $v = \frac{JSZz}{\sqrt{2}}$.

We express the perturbation $\hat{V}(t)$ in terms of the same bosons. Up to a constant term, this yields

$$\begin{aligned}
\hat{V}(t) &= \eta J f(t) \sum_{\mathbf{r}\in A}\sum_{\delta} p_{\delta}\hat{\mathbf{S}}_{\mathbf{r}} \cdot \hat{\mathbf{S}}_{\mathbf{r}+\delta} \\
&= f(t) \sum_{\mathbf{k}}\left[\delta\omega_{\mathbf{k}}\left(\hat{\alpha}_{\mathbf{k}}^{\dagger}\hat{\alpha}_{\mathbf{k}} + \hat{\beta}_{\mathbf{k}}\hat{\beta}_{\mathbf{k}}^{\dagger}\right) + V_{\mathbf{k}}\hat{\alpha}_{\mathbf{k}}^{\dagger}\hat{\beta}_{-\mathbf{k}}^{\dagger} + V_{\mathbf{k}}^{*}\hat{\beta}_{-\mathbf{k}}\hat{\alpha}_{\mathbf{k}}\right],
\end{aligned} \tag{A.6}$$

where

$$\begin{aligned}
\delta\omega_{\mathbf{k}} &= -\eta JSz \frac{|\gamma_{\mathbf{k}}|}{\varepsilon_{\mathbf{k}}}\mathrm{Re}\left(\Gamma_{\mathbf{k}}e^{-i\phi_{\mathbf{k}}}\right), \\
V_{\mathbf{k}} &= -\eta JSz\left[\frac{1}{\varepsilon_{\mathbf{k}}}\mathrm{Re}\left(\Gamma_{\mathbf{k}}e^{-i\phi_{\mathbf{k}}}\right) + i\,\mathrm{Im}\left(\Gamma_{\mathbf{k}}e^{-i\phi_{\mathbf{k}}}\right)\right], \qquad \Gamma_{\mathbf{k}} = \frac{1}{z}\sum_{\delta} p_{\delta}e^{i\mathbf{k}\cdot\delta}.
\end{aligned} \tag{A.7}$$

We do not apply an Oguchi-like correction to the perturbation. Finally we calculate the spin correlation function (defined in Eq. (21)) in LSWT. Using Eq. (18) yields

$$C(\mathbf{R}, t) = \begin{cases} -4S\dfrac{2}{N}\sum_{\mathbf{k}}\left(n_{\mathbf{k}} + \dfrac{1}{2}\right)T_{\mathbf{k}}(t)\dfrac{|\gamma_{\mathbf{k}}|}{\varepsilon_{\mathbf{k}}}\cos\left(\mathbf{k}\cdot\mathbf{R}\right)\mathrm{Re}\,V_{\mathbf{k}}, & \mathbf{R}\in A, \\[4mm] 4S\dfrac{2}{N}\sum_{\mathbf{k}}\left(n_{\mathbf{k}} + \dfrac{1}{2}\right)T_{\mathbf{k}}(t)\left[\dfrac{1}{\varepsilon_{\mathbf{k}}}\cos\left(\mathbf{k}\cdot\mathbf{R} - \phi_{\mathbf{k}}\right)\mathrm{Re}\,V_{\mathbf{k}}\right. \\[2mm] \qquad\qquad\qquad\qquad \left. + \sin\left(\mathbf{k}\cdot\mathbf{R} - \phi_{\mathbf{k}}\right)\mathrm{Im}\,V_{\mathbf{k}}\right], & \mathbf{R}\in B, \end{cases} \tag{A.8}$$

where

$$T_{\mathbf{k}}(t) = -\mathrm{Im}\int_{-\infty}^{t}dt'\,f(t')e^{-i2\omega_{\mathbf{k}}(t-t')} = \frac{1 - \cos(2\omega_{\mathbf{k}}t)}{2\omega_{\mathbf{k}}}, \tag{A.9}$$

where the last equality holds for a $f(t) = \Theta(t)$. Note that for the square lattice, the sin term in Eq. (A.8) vanishes upon taking the $\mathbf{k}$-sum due to inversion symmetry.

# B  Static perturbation

In this section, we consider a static perturbation of the exchange interactions in the Heisenberg Hamiltonian. We construct the mean-field equations, solve these up to first order in the perturbation strength $\eta$ (expressed in the unperturbed mean-field parameters), and finally relate the result to the dynamical mean-field parameters of the main text. The perturbed Hamiltonian is given by

$$\hat{H} = J\sum_{\mathbf{r}\in A}\sum_{\delta}(1 + \eta p_{\delta})\hat{\mathbf{S}}_{\mathbf{r}} \cdot \hat{\mathbf{S}}_{\mathbf{r}+\delta}. \tag{B.1}$$

Applying the Schwinger boson mapping from Eq. (2), including a constraint term, and transforming to momentum space yields

$$\hat{H} = \lambda \sum_{\mathbf{k},s} \left( \hat{a}_{\mathbf{k},s}^\dagger \hat{a}_{\mathbf{k},s} + \hat{b}_{-\mathbf{k},s} \hat{b}_{-\mathbf{k},s}^\dagger - W_{\mathbf{k}} \hat{a}_{\mathbf{k},s}^\dagger \hat{b}_{-\mathbf{k},s}^\dagger - W_{\mathbf{k}}^* \hat{b}_{-\mathbf{k},s} \hat{a}_{\mathbf{k},s} \right),$$

$$\text{where} \quad W_{\mathbf{k}} = \frac{1}{\lambda} \sum_{\delta} (1 + \eta p_\delta) Q_\delta \, e^{i\mathbf{k}\cdot\delta} \tag{B.2}$$

(with anisotropic $Q_\delta$), which is diagonalized by the Bogoliubov transformation (different from the transformation in Eq. (4))

$$\hat{a}_{\mathbf{k},s} = e^{i\phi_{\mathbf{k}}} \left( c_{\mathbf{k}} \hat{\alpha}_{\mathbf{k},s} + s_{\mathbf{k}} \hat{\beta}_{-\mathbf{k},s}^\dagger \right), \qquad \hat{b}_{-\mathbf{k},s} = c_{\mathbf{k}} \hat{\beta}_{-\mathbf{k},s} + s_{\mathbf{k}} \hat{\alpha}_{\mathbf{k},s}^\dagger, \tag{B.3}$$

where $c_{\mathbf{k}} = \cosh\theta_{\mathbf{k}}, s_{\mathbf{k}} = \sinh\theta_{\mathbf{k}}, \tanh(2\theta_{\mathbf{k}}) = |W_{\mathbf{k}}|$. This yields

$$\hat{H} = \sum_{\mathbf{k},s} \Omega_{\mathbf{k}} \left( \hat{\alpha}_{\mathbf{k},s}^\dagger \hat{\alpha}_{\mathbf{k},s} + \hat{\beta}_{-\mathbf{k},s} \hat{\beta}_{-\mathbf{k},s}^\dagger \right), \qquad \Omega_{\mathbf{k}} = \lambda \sqrt{1 - |W_{\mathbf{k}}|^2}. \tag{B.4}$$

The corresponding self-consistency equations are given by

$$S + \frac{1}{2} = \frac{2}{N} \sum_{\mathbf{k}} \left( n_{\mathbf{k}} + \frac{1}{2} \right) \frac{1}{\sqrt{1 - |W_{\mathbf{k}}|^2}}, \tag{B.5}$$

$$Q_\delta = \frac{2J}{N} \sum_{\mathbf{k}} \left( n_{\mathbf{k}} + \frac{1}{2} \right) e^{-i\mathbf{k}\cdot\delta} \frac{W_{\mathbf{k}}}{\sqrt{1 - |W_{\mathbf{k}}|^2}}. \tag{B.6}$$

Next, we expand these equations in powers of $\eta$ and consider the $\mathcal{O}(\eta)$ terms. Furthermore, we consider only $T = 0$ ($n_{\mathbf{k}} = 0$). The mean-field parameters are expanded in the same way as Eq. (12). Furthermore,

$$W_{\mathbf{k}} = c_0 (\gamma_{\mathbf{k}} + \eta G_{\mathbf{k}}) + \mathcal{O}(\eta^2), \qquad \text{where} \quad G_{\mathbf{k}} = \frac{1}{z} \sum_{\delta} (p_\delta + \Delta Q_\delta - \Delta\lambda) e^{i\mathbf{k}\cdot\delta}. \tag{B.7}$$

At $\mathcal{O}(\eta)$, the self-consistency equations are

$$0 = \frac{2}{N} \sum_{\mathbf{k}} \frac{\text{Re}(\gamma_{\mathbf{k}}^* G_{\mathbf{k}})}{\varepsilon_{\mathbf{k}}^3}, \tag{B.8}$$

$$\Delta Q_\delta = \frac{Jz}{2\lambda_0} \frac{2}{N} \sum_{\mathbf{k}} e^{-i\mathbf{k}\cdot\delta} \left[ \frac{G_{\mathbf{k}}}{\varepsilon_{\mathbf{k}}} + c_0^2 \gamma_{\mathbf{k}} \frac{\text{Re}(\gamma_{\mathbf{k}}^* G_{\mathbf{k}})}{\varepsilon_{\mathbf{k}}^3} \right]. \tag{B.9}$$

This can be simplified by using the identity $\sum_{\mathbf{k}} A_{\mathbf{k}} G_{\mathbf{k}} = (\Delta Q_{\text{av}} - \Delta\lambda) \sum_{\mathbf{k}} A_{\mathbf{k}} \gamma_{\mathbf{k}}$, where $\Delta Q_{\text{av}} = \frac{1}{z} \sum_{\delta} \Delta Q_\delta$, and which holds for $A_{\mathbf{k}}$ which has the symmetry $A_{\mathcal{R}\mathbf{k}} = A_{\mathbf{k}}$, where $\mathcal{R}$ is a 90° (square lattice) or a 120° (honeycomb lattice) rotation matrix. This yields

$$0 = (\Delta Q_{\text{av}} - \Delta\lambda) \frac{2}{N} \sum_{\mathbf{k}} \frac{|\gamma_{\mathbf{k}}|^2}{\varepsilon_{\mathbf{k}}^3}, \tag{B.10}$$

$$\Delta Q_{\text{av}} = (\Delta Q_{\text{av}} - \Delta\lambda) \frac{J}{2\lambda_0} \frac{2}{N} \sum_{\mathbf{k}} \frac{|\gamma_{\mathbf{k}}|^2}{\varepsilon_{\mathbf{k}}^3}. \tag{B.11}$$

Combining these two yields $\Delta\lambda = 0, \Delta Q_{\text{av}} = 0$. Next we solve for each individual $\Delta Q_\delta$, using this obtained result. We can write Eq. (B.9) as the matrix equation

$$\Delta Q_\delta = \sum_{\delta'} D_{\delta,\delta'} (p_{\delta'} + \Delta Q_{\delta'}), \tag{B.12}$$

where

$$D_{\delta,\delta'} = \frac{J}{2\lambda_0}\frac{2}{N}\sum_{\mathbf{k}}\frac{1}{\varepsilon_{\mathbf{k}}^3}e^{-i\mathbf{k}\cdot(\delta-\delta')} \qquad \text{(square)},$$

$$D_{\delta,\delta'} = \frac{J}{4\lambda_0}\frac{2}{N}\sum_{\mathbf{k}}\frac{1}{\varepsilon_{\mathbf{k}}^3}\Big[\big(1+\varepsilon_{\mathbf{k}}^2\big)e^{-i\mathbf{k}\cdot(\delta-\delta')} + \big(1-\varepsilon_{\mathbf{k}}^2\big)e^{i2\phi_{\mathbf{k}}}e^{-i\mathbf{k}\cdot(\delta+\delta')}\Big] \quad \text{(honeycomb)}. \tag{B.13}$$

In the basis $\{\delta_1, \delta_2, \dots \delta_z\}$, $\mathbf{D}$ has the same matrix structure of Eq. (C.4) (without the $i$ index). This is a circulant matrix, whose eigenvectors do not depend on the values of $a, b, c, v, w$. We will now show for both the square and honeycomb lattice that $\mathbf{p}$ is an eigenvector of $\mathbf{D}$, i.e. $\mathbf{Dp} = \mathcal{D}\mathbf{p}$. For the square lattice, we can write $\mathbf{p} = \cos(2\phi)(1, -1, 1, -1)$, which is indeed an eigenvector of $\mathbf{D}$. For the honeycomb lattice, we can write $\mathbf{p} = -\frac{1}{2}\cos(2\phi)(1, 1, -2) + \frac{\sqrt{3}}{2}\sin(2\phi)(1, -1, 0)$. Both $(1, 1, -2)$ and $(1, -1, 0)$ are eigenvectors of $\mathbf{D}$. They have the same degenerate eigenvalue. The corresponding eigenvalue is

$$\mathcal{D} = a + b - 2c = \frac{J}{2\lambda_0}\frac{2}{N}\sum_{\mathbf{k}}\frac{1}{\varepsilon_{\mathbf{k}}^3}\big[\cos(k_x) - \cos(k_y)\big]^2 \qquad = 0.2512 \quad \text{(square)},$$

$$\mathcal{D} = v - w = \frac{J}{4\lambda_0}\frac{2}{N}\sum_{\mathbf{k}}\frac{1}{\varepsilon_{\mathbf{k}}^3}\Big\{\big(1+\varepsilon_{\mathbf{k}}^2\big)\big[1 - \cos(\mathbf{k}\cdot\mathbf{a}_1)\big] \\ + \big(1-\varepsilon_{\mathbf{k}}^2\big)e^{i2\phi_{\mathbf{k}}}\big(\gamma_{2\mathbf{k}}^* - \gamma_{\mathbf{k}}\big)\Big\} \qquad = 0.3530 \quad \text{(honeycomb)}. \tag{B.14}$$

Eq. (B.12) can be solved to find

$$\Delta\mathbf{Q} = (1-\mathbf{D})^{-1}\mathbf{Dp} = \frac{\mathcal{D}}{1-\mathcal{D}}\mathbf{p}. \tag{B.15}$$

Finally, we relate this result to the dynamical mean-field parameter $\Delta Q_\delta(t)$ of Eq. (C.8). We can use the final value theorem of Laplace theory to find an analytical expression for the value of the bond parameter in the limit $t \to \infty$.

$$\lim_{t\to\infty}\Delta Q_\delta(t) = \lim_{s\to 0} s\,\Delta\tilde{Q}_\delta(s) = p_\delta\frac{A(0)+B(0)}{1-A(0)-B(0)}, \tag{B.16}$$

and we can see from Eq. (C.5) that $A(0)+B(0) = \mathcal{D}$. Therefore, the limiting value of the bond parameter after a step function perturbation is the same as its value for a static perturbation.

## C  Time-dependent perturbation

In this section, we consider the time-dependent perturbation of the exchange interactions in the Heisenberg Hamiltonian. The analytical solution of the TDSC equations is similar to the calculation of the mean-field parameters for a static perturbation (see App. (B)). At $\mathcal{O}(\eta)$, the TDSC equation (derived from Eq. (19) of the main text) is given by

$$\eta Q_0\Delta Q_\delta(t) = 2J\frac{2}{N}\sum_{\mathbf{k}}\Big(n_{\mathbf{k}} + \frac{1}{2}\Big)e^{-i\mathbf{k}\cdot\delta}e^{i\phi_{\mathbf{k}}}\Big\{\frac{1}{\varepsilon_{\mathbf{k}}}\text{Im}\Big[\int_{-\infty}^t dt' V_{\mathbf{k}}(t')e^{-i2\omega_{\mathbf{k}}(t-t')}\Big] \\ -i\,\text{Re}\Big[\int_{-\infty}^t dt' V_{\mathbf{k}}(t')e^{-i2\omega_{\mathbf{k}}(t-t')}\Big]\Big\}. \tag{C.1}$$

By defining a reduced Laplace variable $\bar{s} = s/2\lambda_0$, this can be written in the Laplace domain as the linear equation

$$\Delta\tilde{Q}_\delta(s) = \sum_{\delta'}D_1(\delta, \delta')(s)\Big[p_{\delta'}\tilde{f}(s) + \Delta\tilde{Q}_{\delta'}(s)\Big] + \sum_{\delta'}D_2(\delta, \delta')(s)\Big[p_{\delta'}\tilde{f}(s) + \Delta\tilde{Q}_{\delta'}^*(s)\Big], \tag{C.2}$$

in terms of the matrices

$$D_1(\boldsymbol{\delta},\boldsymbol{\delta}')(s) = \frac{J}{2\lambda_0}\frac{2}{N}\sum_{\mathbf{k}}\left(n_{\mathbf{k}}+\frac{1}{2}\right)e^{-i\mathbf{k}\cdot(\delta-\delta')}\frac{1}{\varepsilon_{\mathbf{k}}}\left[\frac{(1+i\bar{s})^2}{\varepsilon_{\mathbf{k}}^2+\bar{s}^2}+1\right],$$

$$D_2(\boldsymbol{\delta},\boldsymbol{\delta}')(s) = \frac{J}{2\lambda_0}\frac{2}{N}\sum_{\mathbf{k}}\left(n_{\mathbf{k}}+\frac{1}{2}\right)e^{-i\mathbf{k}\cdot(\delta+\delta')}e^{i2\phi_{\mathbf{k}}}\frac{1}{\varepsilon_{\mathbf{k}}}\left[\frac{1+\bar{s}^2}{\varepsilon_{\mathbf{k}}^2+\bar{s}^2}-1\right], \tag{C.3}$$

whose functional form can be derived from $V_{\mathbf{k}}$. These matrices have a dependency on the nearest neighbour vectors $\boldsymbol{\delta}$, which relates to the anisotropy of the perturbation. To solve the TDSC equation, it is convenient to work in the basis $\{\boldsymbol{\delta}_1,\boldsymbol{\delta}_2,\dots\boldsymbol{\delta}_z\}$. Here, $D_i(\boldsymbol{\delta},\boldsymbol{\delta}')$ (where $i=1,2$) has the matrix structure

$$\text{Square: } \mathbf{D}_i = \begin{pmatrix} a_i & c_i & b_i & c_i \\ c_i & a_i & c_i & b_i \\ b_i & c_i & a_i & c_i \\ c_i & b_i & c_i & a_i \end{pmatrix}, \qquad \text{Honeycomb: } \mathbf{D}_i = \begin{pmatrix} v_i & w_i & w_i \\ w_i & v_i & w_i \\ w_i & w_i & v_i \end{pmatrix}. \tag{C.4}$$

These are circulant matrices, implying that $\mathbf{D}_1$ and $\mathbf{D}_2$ commute and have the same set of eigenvectors. One can prove that $\mathbf{p}$ (the vector $p_{\boldsymbol{\delta}}$ from Eq. (11)) is one of these eigenvectors: $\mathbf{D}_i\mathbf{p} = d_i\mathbf{p}$ (see App. (B)). The corresponding eigenvalues are given by $d_1 = A + iC$, $d_2 = B$, where

$$A(s) = (1-\bar{s}^2)K_1(s)+P_1, \qquad B(s) = (1+\bar{s}^2)K_2(s)-P_2, \qquad C(s) = 2\bar{s}K_1(s),$$

$$K_i(s) = \frac{J}{2\lambda_0}\frac{2}{N}\sum_{\mathbf{k}}\left(n_{\mathbf{k}}+\frac{1}{2}\right)\frac{\tau_{\mathbf{k},i}}{\varepsilon_{\mathbf{k}}}\frac{1}{\varepsilon_{\mathbf{k}}^2+\bar{s}^2}, \qquad P_i = \frac{J}{2\lambda_0}\frac{2}{N}\sum_{\mathbf{k}}\left(n_{\mathbf{k}}+\frac{1}{2}\right)\frac{\tau_{\mathbf{k},i}}{\varepsilon_{\mathbf{k}}},$$

$$\text{Square: } \tau_{\mathbf{k},1} = \tau_{\mathbf{k},2} = (\cos k_x - \cos k_y)^2,$$

$$\text{Honeycomb: } \tau_{\mathbf{k},1} = 1-\cos(\mathbf{k}\cdot\mathbf{a}_1), \qquad \tau_{\mathbf{k},2} = e^{i2\phi_{\mathbf{k}}}(\gamma_{2\mathbf{k}}^*-\gamma_{\mathbf{k}}). \tag{C.5}$$

Eq. (C.2) can be written as the matrix equation

$$\begin{pmatrix} 1-\mathbf{D}_1 & -\mathbf{D}_2 \\ -\mathbf{D}_2 & 1-\mathbf{D}_1^{\dagger} \end{pmatrix}\begin{pmatrix} \Delta\tilde{\mathbf{Q}} \\ \Delta\tilde{\mathbf{Q}}^* \end{pmatrix} = \tilde{f}(s)\begin{pmatrix} \mathbf{D}_1+\mathbf{D}_2 \\ \mathbf{D}_1^{\dagger}+\mathbf{D}_2 \end{pmatrix}\mathbf{p}, \tag{C.6}$$

which has the solution

$$\Delta\tilde{\mathbf{Q}} = \tilde{f}(s)\left[\left(1-\mathbf{D}_1\right)\left(1-\mathbf{D}_1^{\dagger}\right)-\mathbf{D}_2^2\right]^{-1}\left[\left(1-\mathbf{D}_1^{\dagger}\right)\left(\mathbf{D}_1+\mathbf{D}_2\right)+\mathbf{D}_2\left(\mathbf{D}_1^{\dagger}+\mathbf{D}_2\right)\right]\mathbf{p}. \tag{C.7}$$

Note that we now use vectors and matrices from the vector space spanned by $\{\boldsymbol{\delta}_1,\boldsymbol{\delta}_2,\dots\boldsymbol{\delta}_d\}$ inside a matrix equation in a different vector space. Using the aforementioned eigenvalues, this equation can be simplified to

$$\Delta\tilde{Q}_{\boldsymbol{\delta}}(s) = p_{\boldsymbol{\delta}}\tilde{f}(s)\tilde{q}(s), \qquad \tilde{q}(s) = \frac{(1-A+B)(A+B)-C^2+iC}{(1-A+B)(1-A-B)+C^2}. \tag{C.8}$$

Because $\Delta\lambda = 0$, this is already the full solution to the TSC equation.

Combining Eq. (C.8) and Eq. (15) of the main text yields an expression

$$\tilde{V}_{\mathbf{k}}(s) = -\eta z Q_0\left\{\frac{1}{\varepsilon_{\mathbf{k}}}\text{Re}\left[\tilde{f}(s)\left[1+\tilde{q}(s)\right]\Gamma_{\mathbf{k}}e^{-i\phi_{\mathbf{k}}}\right]+i\,\text{Im}\left[\tilde{f}(s)\left[1+\tilde{q}(s)\right]\Gamma_{\mathbf{k}}e^{-i\phi_{\mathbf{k}}}\right]\right\}, \tag{C.9}$$

which can be used in further calculations in combination with Eq. (18) to calculate the dynamics of observables. Note that with this notation for the real and imaginary part of functions in Laplace space, we mean $\text{Re}\,\tilde{g}(s) = \mathcal{L}^{-1}\{\text{Re}\,g(t)\}$ and not $\text{Re}\,\tilde{g}(s) = \text{Re}\,\mathcal{L}^{-1}\{g(t)\}$.

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
