# Peer review of "Time-dependent Schwinger boson mean-field theory of supermagnonic propagation in 2D antiferromagnets"

_SciPost Physics, doi:SciPost Phys. 17, 159 (2024)_

## Round 1 · Referee Report · Anonymous (Referee 1) · 2023-10-23

Report

The paper presents a comprehensive and theoretically well-founded study of supermagnonic propagation in 2D antiferromagnets. The authors use the time-dependent Schwinger boson mean-field theory (SBMFT) to study the dynamics of spin correlations in this context. The paper is well-structured, and it addresses an important and timely topic in the field of magnonics and spintronics. The manuscript provides a valuable contribution to the understanding of magnon propagation and supermagnonic effects which is an important question in the field of magnonics and ultrafast spintronics - the propagation of coherent magnons with short wavelengths.

The paper offers a thorough analysis of supermagnonic propagation, considering quantum effects and magnon-magnon interactions. The use of Schwinger boson mean-field theory to study the space-time dynamics of spin correlations is a novel approach that provides valuable insights.

The authors have successfully developed a methodology for studying the space-time dynamics of spin correlations within the framework of time-dependent Schwinger-boson mean-field theory (SBMFT) in the linear-response regime. This approach allowed them to explore quantum effects arising from both the discrete lattice structure and the quantum nature of the spins in a unified framework.

One of the central aspects of this study was the investigation of the supermagnonic effect, where spin correlations were found to transiently propagate faster than the highest magnon group and phase velocity. The paper compared the propagation patterns in SBMFT and linear spin-wave theory (LSWT) and found that, while the qualitative features were similar, the propagation velocities at short length and time scales exhibited significant differences.
The study systematically examined the square lattice and honeycomb lattice and varied the quantum spin number (S) to gain a deeper understanding of the supermagnonic effect. It was revealed that the effect is intricately tied to the interplay between propagating magnon pairs, a feature shared with LSWT, and a quasi-bound state resulting from magnon-magnon interactions. The strength of this effect is highly dependent on the quantum nature of the spins, the coordination, and the lattice geometry.

This research could lead to the enhancement of the supermagnonic regime and have implications for experimental detection.

Two minor observations are in order, that the authors may take into consideration:

1) Providing a more accessible explanation of the theory, equations, and their physical interpretations would be helpful for readers who may not be familiar with SBMFT.

2) A more detailed comparison and discussion of how their results align or differ from LSWT could provide additional insights into the significance of their findings.

In conclusion, the paper offers valuable insights into the supermagnonic effect, providing a foundation for future studies and potential applications in the field of magnonics and ultrafast spintronics. Henceforth I recommend publication in SciPost Physics.

Requested changes

No changes are requested

  • validity: high
  • significance: high
  • originality: high
  • clarity: high
  • formatting: good
  • grammar: good

Author:  Johan Mentink  on 2024-10-05  [id 4839]

(in reply to Report 1 on 2023-10-23)

We thank the referee for the positive assessment of our manuscript. To make the physics resulting from the SBMFT theory more accessible, we decided to focus on the qualitatively new results that are captured beyond LSWT. Hence we have moved Eqs. (20-27) of Sec. 3.1 to an appendix since they comprise merely technical derivations (see also the feedback from referee 2) .

We have further highlighted the physical effects by adding a paragraph at the end of Sec. 3.1 to explain the difference with LSWT. Unlike LSWT, the SBMFT includes magnon-magnon interactions. When taking these interactions into account, the two-magnon spectrum changes qualitatively: instead of being a (renormalized) sum of non-interacting magnons, the excitation of interacting magnon-pairs causes the emergence of a quasi-bound state. These changes are captured in SBMFT by the time-dependent correction to the bond order parameter, but are missing in LSWT.

---

## Round 1 · Referee Report · Anonymous (Referee 2) · 2023-10-26

Strengths

  • well written and nice figures/illustrations
  • results and methodology seem sound and are clearly stated
  • problem that is addressed is timely and interesting

Report

In the manuscript entitled “Time-dependent Schwinger boson mean-field theory of supermagnonic propogation in 2D antiferromagnets”, Bouman and Mentink study time-dependent Schwinger-boson mean-field theory (SBMFT) and compare it with linear spin-wave theory. The goal is to develop a detailed understanding of what the authors dubbed (in a previous work) the “supermagnonic effect”, i.e., the fact that spin correlations can propagate faster than the highest magnon velocity. Antiferromagnetic models on two different lattices, the square and honeycomb lattice, are studied in parallel, which helps illustrating the concepts and phenomena.

A better understanding and illustration of the “supermagnonic effect” is clearly interesting and timely; although the work appears to be a follow-up of a previous PRL (Ref. 26 in the manuscript) of the authors with more details, it does seem to contain enough new results and explanations to justify publication in a journal like SciPost Physics. However, I would like to ask the authors to consider the following comments:

Requested changes

1) As mentioned above, it is clear that the work is a follow-up to the previous PRL, which is of course completely fine. However, I think the current introduction does not make very clear what the additional aspects/results/insights in the manuscript at hand are. So it seems important to very clearly distinguish it from the previous work. I understand there is already a paragraph (bottom of page 2, top of page 3) talking about this, but it was not clear to me, e.g., to what extend the SBMFT has already been studied in the previous PRL. Things become clear after reading the PRL, but this should not be necessary.
2) The part of the paper from Eq. (20) to Eq. (27) contains a lot of equations and not too much physics. The authors might want to think about moving part of it to an Appendix or at least cutting a few equations out of the draft.

Finally, I notice the typo “nog longer” on page 2.

  • validity: high
  • significance: high
  • originality: good
  • clarity: top
  • formatting: excellent
  • grammar: excellent

Author:  Johan Mentink  on 2024-10-05  [id 4838]

(in reply to Report 2 on 2023-10-26)

Answer 2-1
We thanks the referee for the positive assessment of our research.

In the previous PRL, we already identified that the origin must be quantum fluctuations by studying the dependence on the spin value S using SBMFT. Here, we study more systematically the dependence on quantum fluctuations with the same SBMFT by investigating not only the dependence on S but also the dependence on the lattice coordination z. This gap making is now made explicitly in the introduction text.

Answer 2-2
We agree with the referee that Eqs. (20-27) do not contain key physical insight themselves. They comprise merely the result of technical derivations and convenient notation to derive the results analytically. Therefore, we followed the advice of the referee to move these results to the appendix.

We corrected the typo on page 2.

---

## Round 1 · Referee Report · Anonymous (Referee 3) · 2023-11-8

Report

This is an interesting theoretical work on a topical subject in the field of magnetism. For the most part, the authors do a good job motivating and presenting details of their calculations. Still, in my view, the second part devoted to the time-dependent SBMFT lacks proper physical introduction. In the beginning of Sec. 3 the authors give few references to the prior publications, which may discuss, the light-matter Hamiltonian, but they do not sum up their results and do not provide explanations on writing the time-dependent Hamiltonian (10). In particular, what are typical f(t) and p(delta) relevant to the cited experiments.
The lack of the physical insight into the origin of (10) leaves me puzzled with the subsequent procedure adopted by the authors in their calculations. Namely, the recipe that the self-consistent solution needs to be obtained for each moment of time 't'. This implies that f(t) in (10) is a slowly varying function of time, right? Subsequently, in Sec. 3.2, a step-like f(t) is used. Such a time-dependent perturbation can be presumably realised for quantum gases in optical lattices. Can it be relevant to magnetic solids?

I think the paper deserves publication once the above questions are properly addressed in the text.
  • validity: good
  • significance: high
  • originality: high
  • clarity: good
  • formatting: good
  • grammar: reasonable

Author:  Johan Mentink  on 2024-10-05  [id 4837]

(in reply to Report 3 on 2023-11-08)

We thank the referee for the positive assessment of our paper. To better assess the physical origin of equation (10), we now started with introducing the general light-matter hamiltonian and explicitly review previous publications that have derived them. Subsequently, we specialize to the formulation given in Eq.(10), which is merely a technical convenience for the systematic derivation of the linear-response SBMFT.

In addition, in the same introduction paragraph of Sec. 3, we elaborated on the physical motivation for the form of the function f(t): for solids, the most relevant function for f(t) is a function which mimicks time-envelope of a femtosecond laser pulse (eg a Gaussian in time). Since the laser pulse can perturb the exchange interaction only modestly (at most by a few percent of the unperturbed exchange interaction), we focus on the derivation in the linear-response regime.

The theoretical linear response formalism itself would be valid for any shape of the function f(t). For laser-induced spin dynamics, we are mostly interested in impulsive perturbations. These are characterized by a pulsewidth $\tau$ that is much shorter than the subsequent time scale of the induced dynamics of the system. For the present case, in which the response features oscillations with maximum frequency $\omega_{max}$, this comprises the regime $\tau \omega_{max}\ll 1$.

In numerical simulations, impulsive perturbations can be conveniently implemented by a square pulse $f(t)=1$ for $0< t \leq \tau$. Below we are interested in the short-time dynamics for which the supermagnonic effect is observed. In this regime, the dynamics induced by a single step function $f(t)=\Theta(t)$ was found to closely resemble the square pulse protocol adopted in an earlier publication [G. Fabiani and J.H. Mentink, SciPost Phys. 7, 004 (2019)]. Therefore the step function is used to investigate the supermagnonic effect.

---

## Round 2 · Referee Report · Anonymous (Referee 4) · 2024-10-8

Strengths

See previous report.

Weaknesses

Have been addressed.

Report

I am satisfied by the authors' response and changes in the manuscript, which can be accepted in its current form.

Recommendation

Publish (meets expectations and criteria for this Journal)

---

## Round 2 · Referee Report · Anonymous (Referee 5) · 2024-10-29

Report

In the new submission of their paper the authors have tried to improve on the presentation of their theory. Unfortunately, they have not supplied a list of the essential changes into the manuscript, which complicates significantly a refereeing job. I find that my recommendation to better expose the physical context has not been properly addressed. Still taking in account novelty and general interest I can recommend to publish the paper in the present form.
The only final request is to correct a sentence preceding Eq.(10), which makes no sense to me.

Recommendation

Publish (meets expectations and criteria for this Journal)

---

## Round 2 · List of Changes

- Added explanation on light-matter interaction on relevant pulse shape for condensed matter systems
- Adapted introduction to make novelty beyond previous work more clear
- Moved technical parts of derivation to appendix
- Added paragraph to highlight physics included in SBMFT that goes beyond LSWT

---

## Editorial Decision

published